# Attacks on Machine-Text Detectors Retain Stylistic Fingerprints

**Rafael A, Rivera Soto** [1 2]   **Barry Chen** [2]   **Nicholas Andrews** [1]

## Abstract

Despite considerable progress in the development of machine-text detectors, the ease with which machine-text can be manipulated to evade detection has led to suggestions that the problem is inherently intractable. In this work, we investigate the limits of such evasion strategies. We demonstrate that while current attacks, ranging from prompt engineering to detector-guided optimization can effectively degrade performance of standard detectors, they fail to erase the underlying stylistic "fingerprints" of machine text. We show that few-shot detectors that utilize the stylistic feature space are robust to these evasion attempts, reliably detecting samples even from models explicitly tuned to prevent detection. This raises the question: does style represent a universal defense against machine-detection attacks? We demonstrate that the answer is "no" by introducing a novel paraphrasing approach that simultaneously optimizes for undetectability and adherence to specific human styles. We show that unlike prior methods, this attack effectively evades all considered detectors, including those that utilize writing style. However, we find that this evasion is not absolute: as the number of documents available for analysis grows, the human and machine distributions become distinguishable again. Overall, our findings suggest that reliable machine-text detection requires moving beyond single-document analysis to multi-document analysis.[1]

## 1. Introduction

Large language models (LLMs) can generate fluent text across various domains. While there are many benign uses of LLMs, such as for writing assistance, they may also be abused (Weidinger et al., 2022; Hazell, 2023). To mitigate potential abuse, several machine-text detection systems have been proposed, including zero-shot methods such as Binoculars, DetectGPT, FastDetectGPT, and DNA-GPT (Hans et al., 2024; Mitchell et al., 2023; Bao et al., 2024; Yang et al., 2024), supervised detectors such as RADAR and ReMoDetect (Hu et al., 2023; Lee et al., 2024), and watermarking approaches (Kirchenbauer et al., 2023; Kuditipudi et al., 2024). However, as evasion attacks increase in sophistication, detecting AI-generated text becomes increasingly challenging, raising concerns about the reliability of existing detection methods.

Recent work has demonstrated that zero-shot and supervised detectors can easily be fooled via various attacks ranging from optimization to paraphrasing (Nicks et al., 2024; Koike et al., 2024; Sadasivan et al., 2025; Krishna et al., 2023). For example, Nicks et al. (2024) show that using a detector's "humanness" score as a reward signal in reinforcement learning can effectively lead LLMs to generate text that fools detectors. However, while these optimization approaches defeat many popular zero-shot and supervised detectors (Ippolito et al., 2020; Mitchell et al., 2023; Bao et al., 2024; Hans et al., 2024; Hu et al., 2023; Lee et al., 2024), we show that they fail to obscure the underlying stylistic fingerprints of the model (§3). Specifically, we find that detectors that utilize neural representations of writing style (Rivera Soto et al., 2024) remain robust to the distribution shifts introduced by various evasion methods (Table 1). This suggests that the features targeted by popular evasion methods are distinct from those indicative of authorship style. Crucially, we find that style-based detectors remain robust even when explicitly targeted by standard preference optimization, suggesting that simply optimizing for "non-machine" features is insufficient to close the stylistic gap.

Is detection using stylistic features inherently robust to attacks? To robustly avoid detection, we argue that one must move beyond generic optimization and explicitly optimize *for* author-specific human writing styles. To validate this hypothesis, we introduce a style-aware paraphraser that, conditioning on a few excerpts of a target author, is capable

---
[1]Johns Hopkins University [2]Lawrence Livermore National Laboratory. Correspondence to: Rafael A. Rivera Soto <rafaelrivera-soto@jhu.edu>.

*Proceedings of the $43^{rd}$ International Conference on Machine Learning*, Seoul, South Korea. PMLR 306, 2026. Copyright 2026 by the author(s).

---
[1]The datasets, method implementations, model checkpoints, are available at: `https://github.com/rrivera1849/style-aware-paraphrasing`

| Evasion Strategy | Detection Performance (AUROC) | | | |
|---|---|---|---|---|
| | FastDetectGPT | Binoculars | SemDetect | **StyleDetect** |
| *Optimization Attacks (DPO Nicks et al. (2024))* | | | | |
| No Attack (Baseline) | 65 | 66 | 67 | **97** |
| Target: FastDetectGPT | 35 | 34 | 66 | **96** |
| Target: StyleDetect | 65 | 66 | 64 | **96** |
| *Post-Hoc Attacks* | | | | |
| No Attack (Baseline) | 71 | 79 | 73 | **98** |
| Adversarial (OUTFOX Koike et al. (2024)) | 64 | 74 | 91 | **99** |
| Paraphrasing (DIPPER Krishna et al. (2023)) | 81 | 83 | 72 | **89** |
| TinyStyler (Horvitz et al., 2024b) | 89 | **92** | 69 | 87 |
| Paraphrasing (GPT-4o-mini §5.1) | 59 | 68 | 75 | **99** |
| Prompting (GPT-4o-mini Style-Aware §5.1) | 69 | 74 | 73 | **98** |
| *Average Across Evasion Strategies* | 66 | 70 | 73 | **95** |

*Table 1.* **The robustness of stylistic features against various evasion attacks.** We evaluate the impact of various evasion strategies on detection performance. While generic attacks successfully degrade the performance of standard detectors (FastDetectGPT and Binoculars), **StyleDetect remains robust** across all attack categories. This indicates that neither optimization nor standard paraphrasing is sufficient to erase the stylistic fingerprints of the model.

of mimicking the authorship style while preserving the original meaning. We train our model in two stages: supervised fine-tuning to learn how to paraphrase in the style of human-written exemplars, and preference optimization (Rafailov et al., 2023) to refine generations for undetectability. We show that when applied iteratively on machine-generated text, our system produces outputs that are indistinguishable from human-written text, even to style-based detectors, when only a single sample is available for detection. However, we demonstrate detection performance is recovered as the number of documents available grows.

**Primary contributions** We demonstrate that the stylistic feature space is robust to various evasion attacks. Specifically, we show that while machine-text can be manipulated to defeat standard detectors, they remain identifiable by style-based detectors, as popular evasion attacks are insufficient to remove stylistic fingerprints (§3). To bridge this gap, we introduce a novel training recipe for a state-of-the-art style-aware paraphraser that mimics human writing style, successfully evading detection in the single-sample regime (§4). Finally, we reveal the limits of this evasion: as the number of aggregated documents available for analysis grows ($N > 1$), the distributions of human and machine text become distinguishable again, suggesting a necessary shift in detection paradigms (§6).

## 2. Preliminaries: The Stylistic Feature Space

Central to our study is the concept of a *style representation* (Wegmann et al., 2022; Rivera-Soto et al., 2021; Patel et al., 2025). Unlike semantic representations, which map text to vectors based on meaning or topic, a style representation is a neural model $f_\theta$ that maps a document $x$ to a fixed-dimensional vector $v = f_\theta(x)$ such that the dis-

tance between vectors corresponds to authorship similarity rather than semantic content. Formally, if documents $x_i$ and $x_j$ share the same author but differ in topic, their representations $v_i$ and $v_j$ should exhibit high cosine similarity. These models are typically trained via contrastive learning objectives on large-scale authorship verification tasks, optimizing the embedding space to cluster documents by author regardless of their underlying meaning. Crucially, these representations are often trained on *low-resource* human authors (e.g., those with fewer than 100 documents), and have recently been found to be discriminative of different LLMs (Rivera Soto et al., 2024).

## 3. The Robustness of Stylistic Features Against Attacks

In this section, we investigate whether detectors that utilize the stylistic feature space remain robust against a broad spectrum of evasion strategies. We categorize these strategies into two groups: *optimization attacks*, where the generator is post-trained to evade detection, and *post-hoc attacks*, where machine-text is modified via various strategies.

Following Nicks et al. (2024), we optimize LLMs using Direct Preference Optimization (DPO) (Rafailov et al., 2023). We generate response pairs for every prompt and select the one rated as "more human-like" by a target detector as the preferred sample. We further evaluate robustness against methods that do not require fine-tuning: (1) Paraphrasing using DIPPER (Krishna et al., 2023) and GPT-4o-mini; (2) Adversarial Prompting using OUTFOX (Koike et al., 2024), a prompting framework which uses a detector's prediction labels as examples for in-context learning, (3) Style-Aware Prompting, where we instruct GPT-4o-mini to re-write text

in the style of a specific author using in-context learning, and (4) TinyStyler (Horvitz et al., 2024b), a style transfer method that rewrites text to match a target stylistic exemplar. (see §5.1 for more information about these attacks).

We conduct our evaluations on a held-out test set of Reddit comments using two distinct experimental protocols corresponding to the attack categories. For the optimization attacks, we evaluate across three open-weight model families (Mistral-7B-Instruct, Qwen-2.5-7B-Instruct, and Mistral-Nemo-Instruct) and report the average detection performance. In this setting, the DPO variants are compared directly against their respective base unoptimized checkpoints. For the post-hoc attacks (Paraphrasing, OUTFOX, Style-Aware Prompting), we evaluate on a diverse set of models comprising Mistral-7B-Instruct, Llama-3-8B-Instruct, and GPT-4o-mini. We generate responses using these models and then apply the respective transformation to the output.

In both protocols, we compare standard zero-shot detectors (FastDetectGPT (Bao et al., 2024), Binoculars (Hans et al., 2024)) against the few-shot method, StyleDetect. Crucially, we populate the StyleDetect support set with 100 examples from the *original, unoptimized* base LLM (e.g., vanilla Mistral-7B-Instruct). We acknowledge that this provides StyleDetect with information unavailable to the zero-shot baselines (access to the target distribution). However, this setup is necessary to isolate our primary research question: **Do these attacks successfully erase the stylistic signature of the base model?** If an attack truly altered the underlying authorship style, the similarity between the attacked text and the unoptimized anchors would degrade, causing detection to fail. Thus, high performance in this setting serves as evidence that the attacks fail to bridge the stylistic gap, even if they successfully obfuscate the statistical artifacts relied upon by zero-shot methods. To isolate the robustness of the stylistic feature space from the inherent advantages of the few-shot setting, we introduce SemDetect as a semantic baseline. Equipped with the same 100 exemplars from the unoptimized base LLM, SemDetect employs a semantic encoder to compute the cosine similarity between a test sample and the average embedding (centroid) of the support set.

As shown in Table 1, generic attacks successfully lower the performance of standard zero-shot detectors. For example, optimizing against FastDetectGPT drops its detection AUROC from 65 to 35. Similarly, paraphrasing and adversarial attacks significantly reduce the detection rates of FastDetectGPT and Binoculars. In stark contrast, **StyleDetect remains robust across all attack categories**. Crucially, this performance is not merely an artifact of the few-shot setting: SemDetect, which has access to the same target exemplars but utilizes semantic representations, fails to match this robustness. Interestingly, TinyStyler achieves the most reduction in StyleDetect performance (AUROC drops to 0.87),

indicating some success in altering the stylistic signature. However, this comes at a steep cost: the transformation introduces statistical artifacts that make the text *easier* to detect by zero-shot methods, with Binoculars' performance rising from 79 to 92. This reveals a critical trade-off: existing style transfer methods may disrupt the original style but fail to maintain the properties required for undetectability by zero-shot detectors.

**Why is the stylistic feature space robust?** We argue that optimizing a system to avoid a generic "machine" signal is an ill-posed problem. Human writing is not a monolith; each author has her own idiosyncratic style. As noted in §2, the representations used by StyleDetect are trained to discriminate between low-resource (100 posts or less) human authors (Rivera-Soto et al., 2021), capturing fine-grained features distinct from broad style categories (e.g., "formal" or "casual"). Without specifying a *specific* human author as a target, generic optimization fails to converge to a valid human stylistic fingerprint. Moreover, as demonstrated by TinyStyler, transferring style is not enough to avoid detection by zero-shot detectors. Consequently, to reliably evade detection, an attacker must simultaneously optimize for undetectability and adherence to specific human authorship styles, a challenge we address in the following section.

## 4. Building a Hard to Detect Style-Aware Paraphraser

**Mimicking human writing styles** Given a machine-generated text sample, our goal is to produce a paraphrase that closely mimics the writing style of a human author. However, parallel data that maps machine-generated text to its human-written paraphrase does not exist. Hence, we first build a paraphraser that, given $M$ in-context pairs of machine-generated paraphrases and their human-written originals, maps a new paraphrase back to its original. Such data can be readily generated, for example, by paraphrasing human-written text with an LLM. Formally, given a dataset of human-written texts $x_i$, their machine-generated paraphrases $p_i$, and their corresponding author labels $a_i$, denoted as $\mathcal{D}_{para} = \{(x_i, p_i, a_i)\}_{i=1}^N$, we instruction-tune (Wei et al., 2022)[2] an LLM to model $p(x_i \mid p_i, C_i)$ where $C_i = \{(x_j, p_j) : a_j = a_i, j \neq i\}$ are exemplars pairs (original and paraphrases) from the same author. In practice, for each human-written text $x_i$ we generate $P$ paraphrases, adding all $P * M$ exemplars to the context. Generating multiple paraphrases per human-written text is an efficient way to increase the number of exemplars without incurring the additional cost of collecting more human-written samples.

**Avoiding machine-text detectors** To ensure that the outputs of the system are hard to detect by machine-text detectors, we further optimize our model using direct preference

---

[2]Instruction can be found in §H.4

| Model | PPL (↑) | Generation |
|---|---|---|
| **Mistral-7B** | 18.96 | I know, right? The suspense was killing me! I was on the edge of my seat waiting for those moments. The dogs were a nice consolation prize, though. Still, I'm hoping for a Ramsay Snow reveal in the future. Fingers crossed! |
| **Detector-Guided DPO** | 67.09 | Wishing for 'Ramsay Snow' mention or Jon's retort, yet appreciating the dogs' presence instead. A testament to George R.R. Martin's captivating narrative. |
| **Ours** | 138.96 | Lol yea it was killing me I was so pumped waiting to see those scenes, dogs as a reward was nice but still want some ramsay snow reveal at some point here's to hoping |

*Table 2.* **Qualitative examples on Reddit.** Tokens are colored by their rank in the gpt2-xl probability distribution: **Green** (Top-10), **Orange** (Top-100), **Purple** (Top-1k), and **Red** (>1k). Text generated by our paraphraser exhibits higher perplexity, indicating it is statistically unlikely under standard LLM distributions. More examples in Appendix G.

optimization (DPO) (Rafailov et al., 2023). To build the preference dataset $\mathcal{D}_{\text{pref}}$, we first train a detector[3] to distinguish between the outputs of our system and human-written text. The detector is trained on a separate dataset $\mathcal{D}_{sup}$ that is created by using our system to paraphrase human-written text in the style of random human authors. For each sample in $\mathcal{D}_{\text{pref}}$, we generate 20 outputs, selecting the most human-like under the aforementioned detector as the preferred generation and a random generation as the less preferred. This encourages the model to generate text that is undetectable by the classifier. Prior work uses DPO to encourage models to produce generations that are undetectable by a zero-shot detector (Nicks et al., 2024), which might not capture all the features that make the generations detectable. In contrast, optimizing against a detector specifically trained to identify our system's generations will capture more of the features that make them identifiable. The hyperparameters used to train our system can be found in Appendix E.

**Inference** To defeat detection, our goal is to paraphrase a *fully* machine-generated sample in the style of a human-author. However, during training, only machine paraphrases of *human text* were observed. This introduces a distribution mismatch, as our system was trained on paraphrases of human-text, which oftentimes contain tokens copied from the original human-text. To bridge this gap, we iteratively apply our style-aware paraphraser, gradually reducing the distributional mismatch. At each iteration, we generate 10 candidates, and choose the top-$P$ (number of paraphrases ingested by our system) that best preserve the semantics of the original text according to SBERT[4] for the next iteration. In the final iteration, we simply pick the candidate that best preserves the meaning of the original text. When our system is applied to paraphrases of human-written text, we simply generate one candidate generation.

**Connection to other paraphrasers** We note, that unlike DIPPER (Krishna et al., 2023), another paraphraser designed for evading machine-text detectors, ours allows

for conditioning on a low-resource authorship style. Prior work of its kind (Horvitz et al., 2024b;a; Khan et al., 2024) focuses on the task of style-transfer, where human-written text is re-written in the style of another human author. Ours is the first that to our knowledge is applied to re-writing machine-generated text. It's also the first paraphraser of its kind that, to our knowledge, includes post-training with DPO for undetectability, achieving a new state-of-the-art in both undetectability and the traditional task of style-transfer (§6.1, Appendix D).

## 5. Experimental Procedure

### 5.1. Baselines

**Prompting (style-aware)** We prompt `gpt-4o-mini` to rewrite machine paraphrases in a given author's style using the same instruction as our system (see Appendix H).

**Paraphrasing** Paraphrasing has been shown to be an effective attack against detectors (Krishna et al., 2023; Sadasivan et al., 2025; Soto et al., 2025), as it alters surface-level features while preserving semantic contents. As such, we evaluate against two paraphrasing baselines. Our first paraphrasing baseline prompts gpt-4o-mini to paraphrase machine-generated text. Our second baseline uses DIPPER (Krishna et al., 2023), an 11 billion parameter paraphrasing model built to evade detectors.

**OUTFOX** is an attack that incorporates in-context examples of text detected as human or machine by a detector, prompting the LLM to generate text that would be detected as human (Koike et al., 2024) by the detector. We chose to include 16 text samples along with the detection results of StyleDetect (instantiated with 100 documents).

**TinyStyler** is a lightweight (800M parameter) style-aware paraphraser trained on Reddit that uses pre-trained author representations for efficient few-shot style transfer (Horvitz et al., 2024b). In contrast, our system tunes a Mistral-7B with LoRA (Hu et al., 2022), does not rely on author representations, and is explicitly optimized to evade machine-text detectors.

---
[3]`FacebookAI/roberta-base`

[4]`sentence-transformers/all-mpnet-base-v2`

**Detector-guided DPO** Following Nicks et al. (2024), we use the "humanness" score from a zero-shot machine-text detector as the reward signal for DPO. Specifically, for each human exemplar in the preference-tuning datasets, we generate two comments, reviews, or blog snippets using Mistral-7B. We then use FastDetectGPT (Bao et al., 2024) to score each comment, selecting the one rated most human-like as the preferred generation.

## 5.2. Datasets

**Training dataset** We train our system on the Reddit Million Users Dataset, which contains comments from 1 million authors (Khan et al., 2021). To ensure that the authors are stylistically diverse while meeting our computational constraints, we further subsample the dataset using stratified sampling in stylistic space. To generate the paraphrases required to train our system, we prompt Mistral-7B-Instruct to produce 5 paraphrases for each comment in the collection just described.

**Preference tuning datasets** For methods that require preference data, namely ours and Detector-Guided DPO, we subsample additional text from each domain, including Reddit, Amazon reviews (Ni et al., 2019), and Blogs (Schler et al., 2006). Specifically, we draw 10,000 samples each from unique authors in the Reddit and Amazon datasets, and 6,000 from the Blogs dataset, ensuring all authors are distinct and disjoint from those in training and evaluation sets. We note that while Detector-Guided DPO utilizes data from all three domains, our method is trained exclusively on the Reddit samples.

**Evaluation data: machine-text detection** We evaluate our approach across three domains: Reddit, Amazon reviews, and Blogs[5]. To generate machine text, we prompt[6] one of Mistral-7B-Instruct, gpt-4o-mini, or Llama-3-8B-Instruct, chosen uniformly at random, to create new comments, reviews, or blog snippets (see prompts in Appendix H)[7]. Crucially, we ensure that the generated texts match the length of the human distribution to rule out length-based artifacts (see Appendix F for details). Each baseline described in §5.1 is then applied to modify this generated text to evade detection. The only exception is Detector-Guided DPO, which generates the text directly, rather than modifying pre-existing outputs. For methods that require target exemplars, including our own, we randomly select an author from the dataset to define the target style and provide 16 of their texts as exemplars.

## 5.3. Metrics and Detectors

**Metrics** To measure the performance of machine-text detectors, we use the standard area under the curve of the receiver operating curve, referred to as AUROC. To better align with real-world scenarios where false-positives are costly, we calculate the partial area for FPRs less than or equal to 1%, which we refer to as AUROC(1). To measure how well the meaning of text is preserved after modification, we use SBERT[8], computing the cosine similarity between embeddings of the original and modified text. Finally, to measure how well the style-aware paraphrasing methods introduce the target style, we use CISR[9], computing the cosine similarity between embeddings of the generated text and target exemplars.

**Detectors** To evaluate how detectable our generations are, we use various detectors, including Rank (Gehrmann et al., 2019), LogRank (Solaiman et al., 2019), FastDetectGPT (Bao et al., 2024), Binoculars (Hans et al., 2024), ReMoDetect (Lee et al., 2024), RADAR (Hu et al., 2023), and StyleDetect (Rivera Soto et al., 2024)[10]. For FastDetectGPT, we use gpt-neo-2.7B, the backbone originally used by the authors. For Rank and LogRank, we use gpt2-xl as the backbone. StyleDetect operates in a few-shot setting, requiring exemplars from the machine-text class; we provide $K = 100$ such examples drawn from random machine-generated text in our dataset that was *not* produced by any of the evaluated methods[11]. We also include two StyleDetect variants that use different style representations: one with CISR embeddings (StyleDetect-CISR) and another with StyleDistance[12] embeddings (StyleDetect-SD). In total, we evaluate nine detectors across trained classifiers (RADAR, ReMoDetect), zero-shot detectors (Rank, LogRank, FastDetectGPT, Binoculars), and few-shot stylistic detectors (StyleDetect, StyleDetect-CISR, StyleDetect-SD).

## 6. Experiments

The goal of our main experimental evaluations is to: (1) demonstrate that our system best evades machine-text detectors §6.1 and (2) show that our approach best closes the gap between human-written and machine-generated styles §6.2.

---

[5]We also report results on essays from the GEDE (Gehring and Paaßen, 2025) dataset in Appendix L, where we also explore two different strategies for handling long documents with our approach.

[6]Using top-p of 0.9 and temperature of 0.7.

[7]We expand this set of models to the newer Qwen3-8B, Qwen3-14B, and Mistral-Nemo in Appendix I, and find that our main conclusions still hold.

[8]sentence-transformers/all-mpnet-base-v2

[9]AnnaWegmann/Style-Embedding

[10]We also explore simple prompting of LLMs for detection in Appendix J, finding them to be poor detectors, thus further motivating the usage of specialized detectors.

[11]We explore the robustness of StyleDetect when provided with support exemplars from *different* than that of the target LLMs in Appendix K, finding that it remains robust even in such settings.

[12]StyleDistance/styledistance

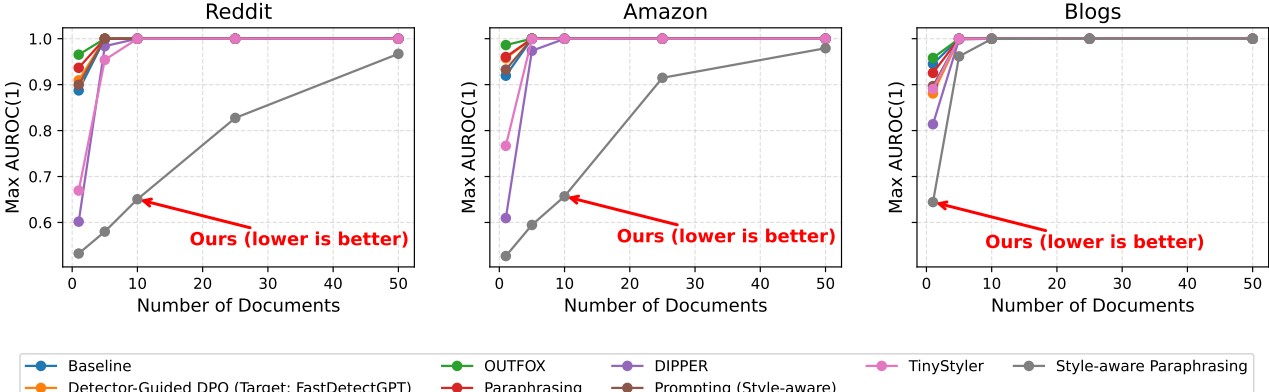

*Figure 1.* Detection performance (AUROC(1), *lower is better*) of the *strongest* detector for each sample size and method combination. **Our detector evasion approach is the least detectable across all three domains, including Amazon and Blogs, which were not seen during training.**

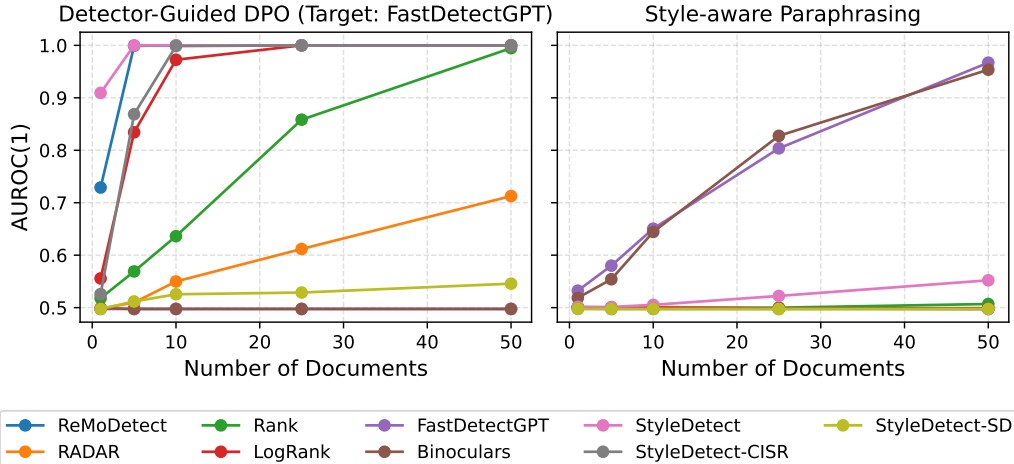

*Figure 2.* Detection performance (AUROC(1), *lower is better for the re-writer*) of various detectors as the sample size increases (left: Detector-Guided DPO, right: Ours). **Our detector evasion approach is consistently harder to detect across all detectors.** Detector-Guided DPO becomes detectable with just 5 samples, while our approach remains robust up to 50. We report the performance of all detectors, evaluated on all methods and all datasets in Appendix A.

| Methods → | Prompting (Style-aware) | Paraphrasing | DIPPER | TinyStyler | Ours |
|---|---|---|---|---|---|
| Edit Distance | 134.05 (81.52) | 156.57 (74.50) | 227.39 (117.94) | 212.58 (101.71) | 199.09 (94.25) |
| Semantic Sim. | 0.91 (0.11) | 0.93 (0.07) | 0.84 (0.11) | 0.78 (0.13) | 0.85 (0.12) |

*Table 3.* Character edit distance, and semantic similarity of the methods that transform text (standard deviation in parenthesis). Results averaged across datasets, for full breakdown see Appendix B. **Our method achieves a high edit distance** (199.09) **comparable to aggressive paraphrasers like DIPPER** (227.39)**, yet maintains a higher semantic fidelity** (0.85) **than competing style-transfer methods like TinyStyler** (0.78)**.**

## 6.1. Machine-Text Detection

In this section, we study whether machine-text detectors are robust against various attacks as the number of documents available grows. Although two distributions may appear indistinguishable on a per-sample basis, their differences become more apparent as the number of samples increases (Chakraborty et al., 2024). For each detector, we compute its score $s_i$ by taking the sample mean of its outputs over $N$ documents. For each value of $N$, we report the *best* score achieved across each of the detectors described in §5 for a *pessimistic* estimate of the detectability of each attack. These results are shown in Figure 1.

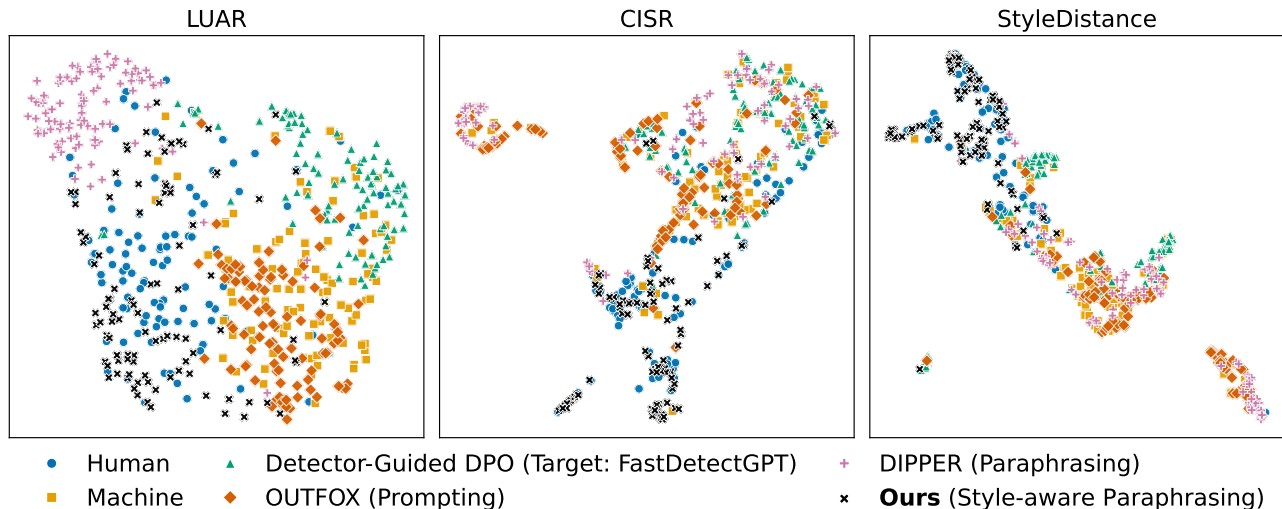

LUAR       CISR       StyleDistance

- Human
- Machine
- ▲ Detector-Guided DPO (Target: FastDetectGPT)
- ◆ OUTFOX (Prompting)
- + DIPPER (Paraphrasing)
- × **Ours** (Style-aware Paraphrasing)

*Figure 3.* UMAP (McInnes et al., 2018) projections of representations that capture writing style for comments in the Reddit domain, using LUAR (Rivera-Soto et al., 2021), CISR (Wegmann et al., 2022), and StyleDistance (Patel et al., 2025). Each point corresponds to a document of at most 128 tokens. **Our style aware paraphraser better closes the gap between human-written and machine-generated text (compare ● with ×).**

| Method | s/sample | Inference Framework |
|---|---|---|
| Detector-Guided DPO (Nicks et al., 2024) | 0.55 | vLLM |
| DIPPER | 1.25 | HuggingFace Transformers |
| TinyStyler | 1.53 | HuggingFace Transformers |
| Ours (total, 3 iterations) | 5.64 | vLLM |

*Table 4.* Per-sample inference time on a single NVIDIA A100 (80GB), batch size 1, averaged over 100 samples after a 5-sample warmup. DIPPER and TinyStyler are not supported by dedicated inference engines such as vLLM.

We find that our approach is the least detectable, even in domains for which it was not trained (Amazon and Blogs). Although our approach transfers well to Amazon, we find that it becomes detectable with just 5 samples in the Blogs domain. We attribute this to the large domain mismatch between the training data (Reddit), favoring informal social media text over the more structured, formal blogs text. To better understand the differences between each detector, we break down the per-detector performance for our method and Detector-Guided DPO on Reddit in Figure 2. The results highlight that although Detector-Guided DPO is robust against FastDetectGPT, the detector it was explicitly optimized against, as well as others that rely on similar token-level features, it remains easily identifiable by StyleDetect, which leverages writing style. In contrast, our approach shows a better trade-off in evading zero-shot detectors (Fast-DetectGPT, Binoculars, Rank, and LogRank) and stylistic detectors (StyleDetect, StyleDetect-CISR, and StyleDetect-SD). Finally, in Table 3, we show the semantic similarity and the character edit distance of each approach that relies on transforming text. We find that our approach preserves the meaning of the original text (similarity $\geq 0.85$), while making on average $+43$ more character edits than regular

paraphrasing. We attribute this increase in edits to the necessary constraint of following the target author's writing style.

We additionally report per-sample inference times in Table 4, measured with a batch size of 1 on a single NVIDIA A100 (80GB) GPU and averaged over 100 samples after a 5-sample warmup. We exclude API-based baselines (OUTFOX, Style-Aware Prompting, and Paraphrasing via `gpt-4o-mini`) since their latency reflects server-side infrastructure rather than the cost of the method itself. Our attack runs three iterations of generation and SBERT re-ranking, each generating 10 candidates, and is consequently slower per sample than the other methods. This overhead is intrinsic to the iterative procedure, and from a defense perspective it raises the cost of large-scale attack deployment.

### 6.2. Visualizing the Space of Writing Styles

We now turn to evaluating whether the approaches considered successfully close the gap between the distributions of human-written and machine-generated writing styles. We choose 100 samples from Reddit generated by each of Detector-Guided DPO, DIPPER, OUTFOX, and our

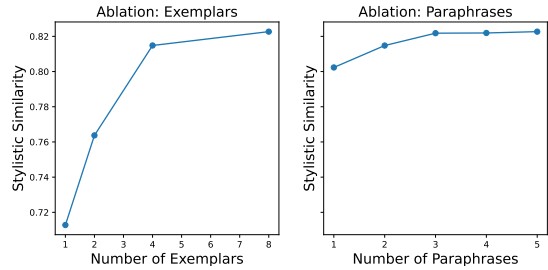

*Figure 4.* Similarity to the target style as a function of $P$ (number of paraphrases per source text, right) and $M$ (number of target exemplars, left). Increasing either $P$ or $M$ consistently improves stylistic similarity.

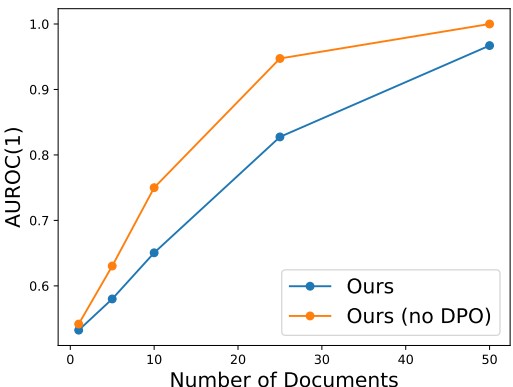

*Figure 5.* Performance of the *best* detector on Reddit for each sample size evaluated on outputs of our style aware paraphraser with, and without DPO. DPO helps maintain the generations undetectable.

style-aware-paraphraser at random. This choice of methods covers the main modalities of detection evasion systems, namely, optimization using DPO, prompting, and paraphrasing. We then embed these generations across three different neural representations of writing-style: LUAR (Rivera-Soto et al., 2021), CISR (Wegmann et al., 2022), and StyleDistance (Patel et al., 2025). We show the results of this in Figure 3. We observe that across all three representations of writing style, our method is qualitatively the one that best closes the gap between human and machine text.

### 6.3. Ablations

In this section, we ablate key hyper-parameters of our system—specifically, $M$, the number of target exemplars provided as context, and $P$, the number of paraphrases generated per exemplar. We show the results in Figure 4, noting that as $M$ or $P$ increases, the stylistic similarity to the target increases. Moreover, we evaluate the worst case detectability as the sample size grows, comparing versions of our system with and without post-training with DPO, finding it to improve the overall performance in Figure 5.

Finally, in Table 5 we quantify how much semantic content is lost across the three iterations of our inference procedure by computing the SBERT cosine similarity between the original machine text and the output of each iteration. Across all three domains, similarity to the original remains above $0.87$ even after three iterations, indicating that the iterative procedure does not introduce substantial semantic drift.

| Stage | Reddit | Amazon | Blogs |
|---|---|---|---|
| Paraphrase | 0.90 (0.09) | 0.96 (0.04) | 0.92 (0.10) |
| Ours (Iteration 1) | 0.93 (0.08) | 0.96 (0.04) | 0.93 (0.11) |
| Ours (Iteration 2) | 0.90 (0.10) | 0.93 (0.05) | 0.90 (0.11) |
| Ours (Iteration 3) | 0.88 (0.11) | 0.92 (0.06) | 0.88 (0.12) |

*Table 5.* SBERT cosine similarity between the original machine text and the output at each stage of our iterative inference procedure (standard deviation in parenthesis). Across all three domains, semantic similarity remains high after three iterations.

## 7. Related Works

**Machine-text detection** Since the advent of LLMs, several lines of research have focused on distinguishing between human-written and machine-generated text. Zero-shot methods (Gehrmann et al., 2019; Ippolito et al., 2020; Bao et al., 2024; Hans et al., 2024) leverage features from the predicted token-wise conditional distributions to separate the distributions. For example, Gehrmann et al. (2019) observes that human-written text tends to be more "surprising," as humans often use tokens that fall into the lower-probability regions of the model's predictive distribution. This observation suggests that humans exhibit personal lexical preferences not easily generated by LLMs. Another line of work relies on supervised detectors (Solaiman et al., 2019; Hu et al., 2023), which have shown strong performance but can be sensitive to distribution shifts at test time. More recently, Rivera Soto et al. (2024) has introduced a detector that uses features indicative of writing style. Finally, watermarking methods (Kirchenbauer et al., 2023; Kuditipudi et al., 2024) introduce detectable biases during generation, though they require the watermarking mechanism to be applied at generation time, an assumption that may not hold in adversarial settings.

**Style-aware paraphrasing** aims to generate paraphrases that reflect a specific target style. Many existing approaches focus on coarse-grained styles, such as formality, informality, Shakespearean English, or poetry (Krishna et al., 2020; Liu and May, 2025), often by training multiple inverse paraphrasing models that transform a neutral version of text into the desired style. Another line of work targets low-resource authorship styles commonly found in social media, using methods such as prompting (Patel et al., 2024), training lightweight models (Horvitz et al., 2024b; Liu et al., 2024), applying diffusion models iteratively (Horvitz et al., 2024a), or using energy-based sampling to optimize for a target

style (Khan et al., 2024). Our approach targets low-resource authors, but further distinguishes itself by not relying on embeddings that capture features indicative of writing style, and by optimizing for undetectability.

**Defeating detectors**    Another line of work aims to defeat machine-text detectors, either through paraphrasing (Krishna et al., 2023; Sadasivan et al., 2025), by prompt optimization (Lu et al.), by adding a single space in the generation (Cai and Cui, 2023), with homoglyphs (Creo and Pudasaini, 2025), or more recently by post-training LLMs with DPO to prefer generations that evade detection (Nicks et al., 2024; Wang et al., 2025). However, we show that these approaches fail to close the gap between human and machine-text distributions, as they primarily manipulate surface-level features without altering the underlying writing style (§3). Our method is the first to jointly optimize *for* author-specific human writing styles and *against* the features exploited by most detectors.

## 8. Conclusion

**Outlook for machine-text detection**    Our findings paint a mixed picture for the feasibility of machine-text detection. On one hand, we expose a key limitation of various evasion approaches, showing that they remain identifiable by detectors that utilize writing style. This initial finding offers a glimmer of hope for machine-text detection. However, we subsequently demonstrate a new attack using style-aware paraphrasing, which is universally effective against all the detectors tested, including those based on writing style. Nonetheless, we show that as the number of documents available for detection grows, there is a point at which the distributions of human and our paraphrased text become separable. Thus, our work suggests a new regime for reliable machine-text detection, where detection decisions about the authenticity of a given source (e.g., author, publication, student, account etc.) must be made based on multiple writing samples, rather than on a document-by-document basis. Particularly in settings such as academia where there may be a large cost to individuals accused of unauthorized AI usage, we argue that this approach is necessary to reduce the incidence of false positives.

**On the practicality of multi-document detection**    A reasonable concern is that the attacker can always provide arbitrarily many writing samples of the target author, while the defender can only see what the user chooses to expose, creating an information asymmetry. We note, however, that in many of the settings where machine-text detection actually matters, the defender already possesses multiple writing samples per author as a byproduct of normal activity. In academic integrity, instructors and institutions retain a student's prior essays, problem-set write-ups, and discussion-board posts; in peer review, program chairs can pool the multiple reviews a single reviewer submits to one venue; in social media and content moderation, platforms such as Reddit, Amazon, and blogs (the three domains studied in this paper) retain users' full post histories. We frame our contribution accordingly: we *characterize* the conditions under which detection is robust, rather than claiming such conditions are universally satisfied.

**Limitations**    While the proposed style-aware paraphraser makes text less detectable, and better closes the distributional gap between human-written and machine-generated text, it has several limitations. First, the approach requires access to exemplars from human authors as demonstrations of diverse writing styles, which might not be available in all scenarios. Second, it necessitates LLM-generated paraphrases, which introduces inference-time costs and can introduce a semantic drift in the generations. Third, the iterative inference time procedure further increases computational costs, making it less suitable for low-compute scenarios. While these are limitations from the perspective of an adversary seeking to *evade* machine-text detection, they may be viewed in positive light from the perspective of machine-text *detection*, as they may place practical limits on the applicability of the attack.

## Impact Statement

The ability to generate convincing machine-generated text poses a significant risk of abuse. This paper contributes an improved understanding of methods to detect machine-generated text, as well as attacks which may hamper the detection of machine-generated text. By studying such attacks, we contribute a better understanding of the limitations of current state-of-the-art defenses, as well as opening the door to future improvements in machine-text detection techniques. Overall, our findings underscore limitations of previous detection regimes, and suggest that certain feature spaces may be inherently more robust for detection.

## Acknowledgements

Part of this work was performed under the auspices of the U.S. Department of Energy by Lawrence Livermore National Laboratory under Contract DEAC52-07NA27344. This work was supported in part by the Office of the Director of National Intelligence (ODNI), Intelligence Advanced Research Projects Activity (IARPA), via the HIATUS Program contract #D2022-2205150003, and by the Johns Hopkins University Data Science and AI Institute. The views and conclusions contained herein are those of the authors and should not be interpreted as necessarily representing the official policies, either expressed or implied, of ODNI, IARPA, the Johns Hopkins University Data Science and AI Institute, or the U.S. Government. The U.S. Government is authorized to reproduce and distribute reprints for governmental purposes notwithstanding any copyright annotation therein.

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

## A. Breakdown of Performance by Method, Dataset, and Detector

Figure 6 and Figure 7 expand the per-detector view in Figure 2 to every attack method. Each panel reports `AUROC(1)` (*lower is better for the attacker*) of all nine detectors of §5.3 on the three domains, as a function of the number of documents per author $N$. The Baseline panel (top-left of Figure 6) serves as the reference: every detector saturates at `AUROC(1)` $\approx 1.0$ with a handful of documents. Comparing the remaining panels to the Baseline shows how each attack suppresses detection: prior attacks (Detector-Guided DPO, Prompting, DIPPER, Paraphrasing, OUTFOX, TinyStyler) degrade some detectors but leave at least one strong detector intact—typically a stylistic one—whereas *Ours* (bottom-right of Figure 7) is the only attack that simultaneously suppresses every detector at small $N$ across all three domains, while detectability still recovers as $N$ grows.

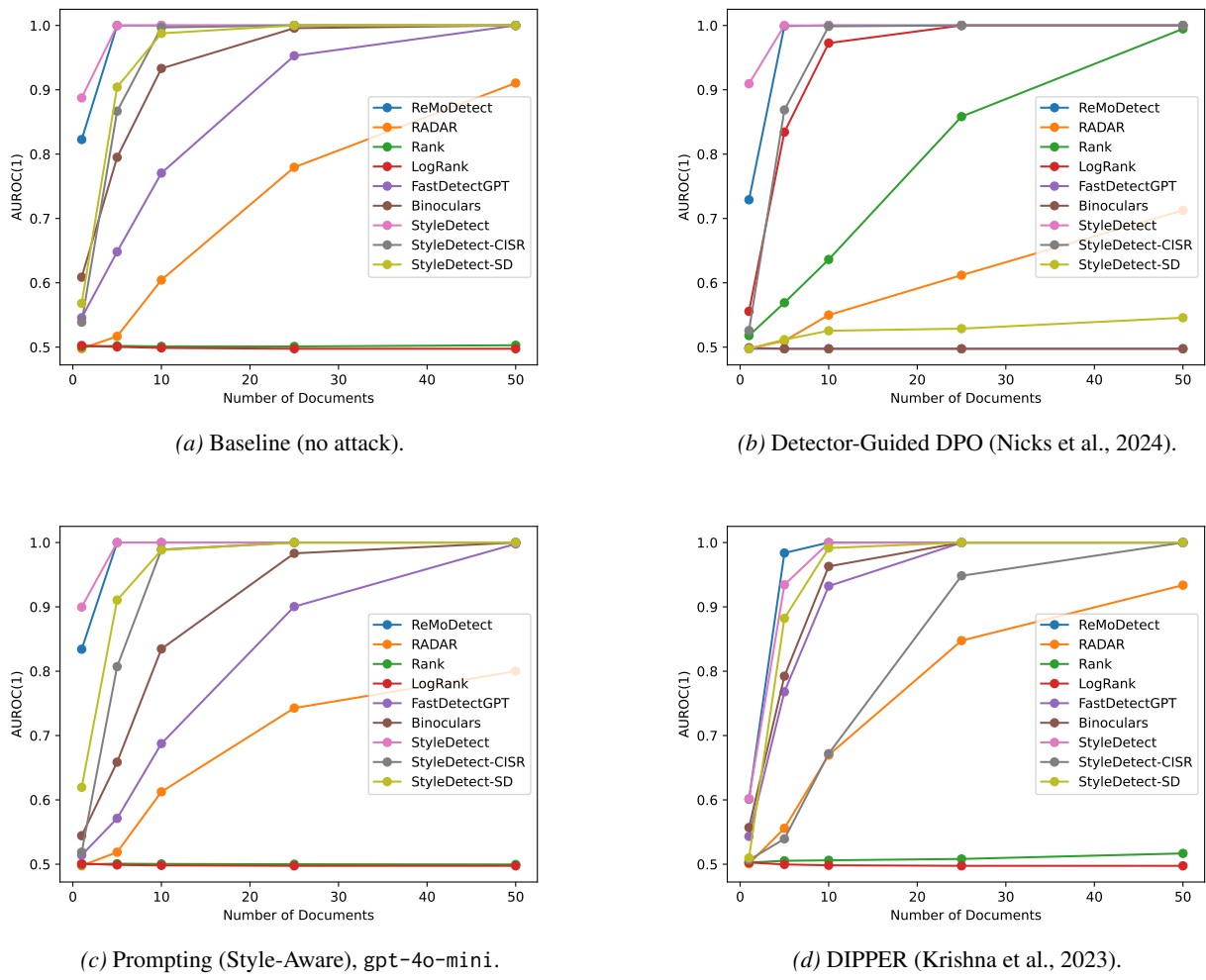

*(a)* Baseline (no attack).

*(b)* Detector-Guided DPO (Nicks et al., 2024).

*(c)* Prompting (Style-Aware), `gpt-4o-mini`.

*(d)* DIPPER (Krishna et al., 2023).

*Figure 6.* Per-detector `AUROC(1)` (*lower is better for the attacker*) on Reddit, Amazon, and Blogs as a function of documents per author $N$, for the Baseline and prior optimization/prompting/paraphrasing attacks. Each line within a panel is one of the nine detectors of §5.3. Compare to the stylistic attacks in Figure 7.

## B. Breakdown of Edit Distance and Semantic Similarity by Dataset

Table 6 expands the cross-domain averages reported in Table 3 of the main text, breaking down character edit distance and SBERT semantic similarity by domain (Reddit, Amazon, Blogs). The trend observed in the main text holds within each domain individually: our method matches aggressive paraphrasers like DIPPER on edit distance while preserving semantics substantially better than the closest style-transfer baseline, TinyStyler.

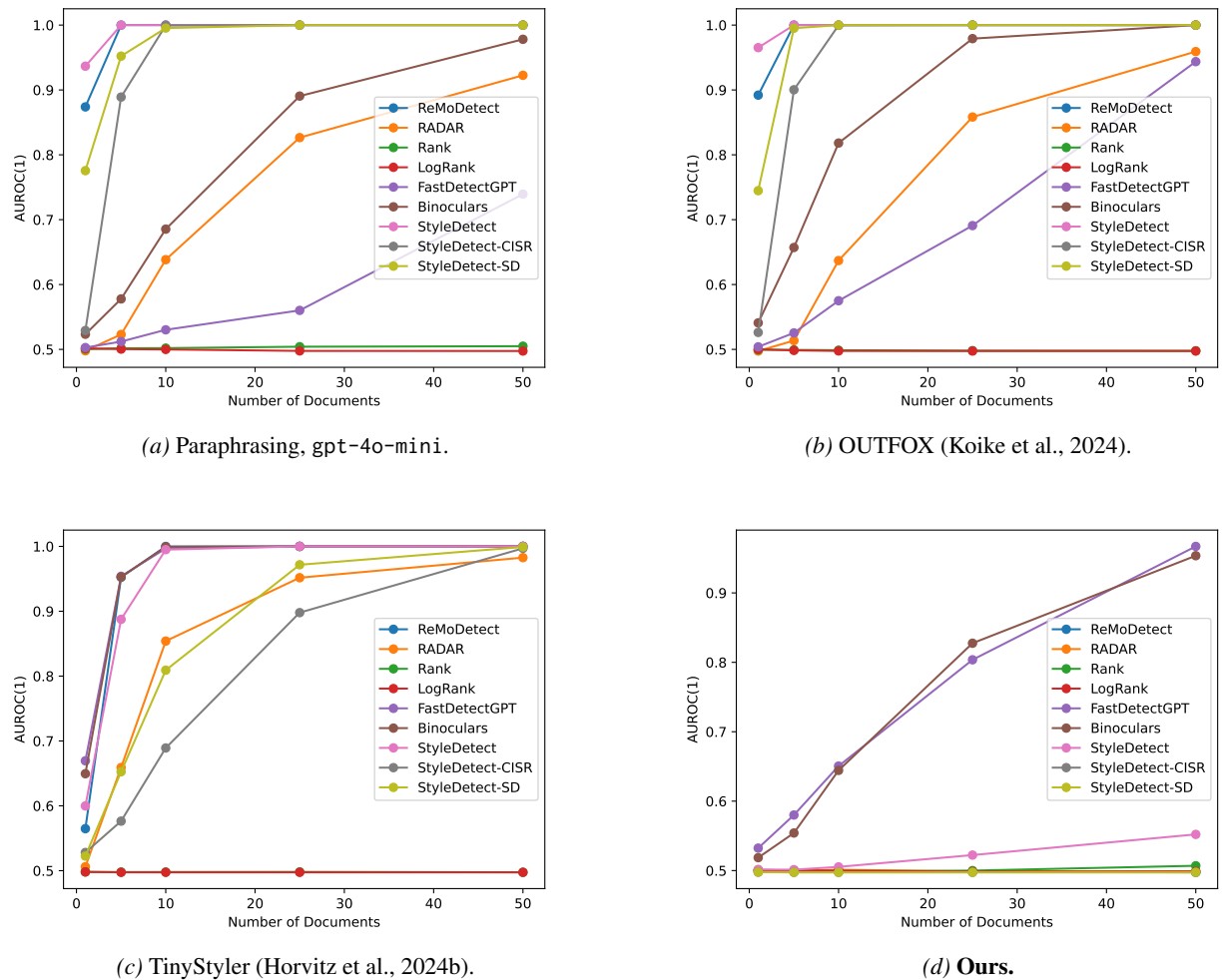

*Figure 7.* Per-detector `AUROC(1)` for the remaining baselines and our method. Prior style-aware attacks (Paraphrasing, OUTFOX, TinyStyler) still leave at least one strong detector intact at small $N$; *Ours* (bottom-right) is the only attack that holds every detector near chance at small $N$ across all three domains, while detectability still recovers as $N$ grows.

| Methods → | Prompting | Paraphrasing | DIPPER | TinyStyler | Ours |
|---|---|---|---|---|---|
| Reddit | | | | | |
| Edit Distance | 107.33 (73.00) | 122.74 (72.97) | 168.02 (94.02) | 158.78 (83.26) | 169.57 (87.90) |
| Semantic Sim. | 0.87 (0.14) | 0.90 (0.09) | 0.76 (0.16) | 0.77 (0.15) | 0.82 (0.15) |
| Amazon | | | | | |
| Edit Distance | 128.06 (76.84) | 143.12 (66.83) | 223.55 (139.83) | 209.61 (110.37) | 178.01 (82.78) |
| Semantic Sim. | 0.94 (0.05) | 0.96 (0.04) | 0.96 (0.04) | 0.84 (0.11) | 0.90 (0.09) |
| Blogs | | | | | |
| Edit Distance | 166.75 (94.71) | 203.85 (83.71) | 290.62 (119.97) | 269.35 (111.50) | 249.68 (112.06) |
| Semantic Sim. | 0.90 (0.14) | 0.92 (0.10) | 0.81 (0.14) | 0.73 (0.14) | 0.85 (0.13) |

*Table 6.* Mean character edit distance, and semantic similarity of the different methods evaluated, broken down by domain (standard deviations in parenthesis). Detector-Guided DPO generates samples from scratch, as opposed to transforming text, therefore there is no reference for comparison.

# C. `AUROC` Performance

In this section, we report the results of the experiment described in §6.1 using the full AUROC (Figure 8).

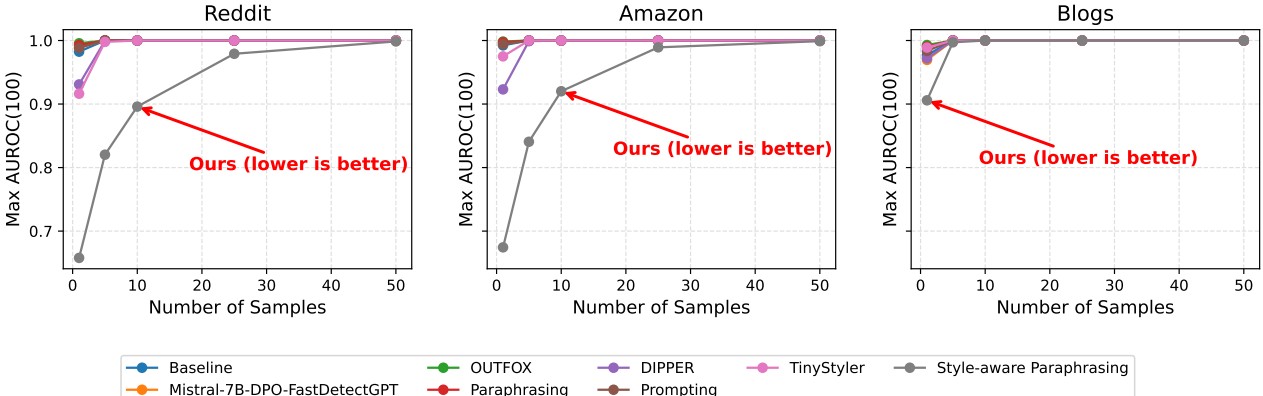

*Figure 8.* Detection performance (AUROC, *lower is better*) of the *strongest* detector for each sample size and method combination. Our detector evasion approach is the least detectable across all three domains, including Amazon and Blogs, which were not seen during training.

## D. Style-Transfer Performance

To benchmark the efficacy of our style-aware paraphraser against existing state-of-the-art methods, we compare its performance to TinyStyler, a recent approach for author-conditioned style transfer.

**Evaluation Dataset**    We constructed a specialized evaluation subset sampled from the Reddit dataset described in §5.2. To ensure robust evaluation across distinct linguistic communities, we sampled $180$ author pairs from four stylistically diverse subreddits: r/WallStreetBets (slang-heavy, financial jargon), r/Australia (regional dialect), r/AskHistorians (formal, academic), and r/news (neutral, journalistic). For each pair, one author serves as the source content provider and the other as the target stylistic exemplar.

**Results**    We evaluate performance on two axes: *stylistic similarity* (measured via CISR (Wegmann et al., 2022) embeddings)[13] and *semantic similarity*[14]. Our approach demonstrates significant improvements over TinyStyler on both metrics. Specifically, we improve stylistic similarity by $+0.12$ (increasing from $0.71$ to $0.83$) while simultaneously enhancing semantic preservation by $+0.09$ (increasing from $0.74$ to $0.83$). This indicates that our method is better able to adopt the target style without sacrificing the underlying content of the message.

## E. Training Hyperparameters and Compute Resources

**Hyper-parameters for experiments in §3**    We optimize each LLM for 3 epochs using DPO with a regularization penalty of $\beta = 0.1$.

**Training hyper-parameters for our style-aware paraphraser**    Our system is parametrized using Mistral-7B, trained for 1 epoch on the Reddit dataset described in §5.2 with a constant learning rate of $2e^{-5}$, using LoRA (Hu et al., 2022) for efficient fine-tuning, setting $r = 32$, $\alpha = 64$, and $d = 0.1$. For the preference-tuning stage, we train our system with $\beta = 5$, and a constant learning rate of $1e^{-6}$.

**Training hyper-parameters for Detector-Guided DPO**    Following (Nicks et al., 2024), we use DPO, training the method for 3 epochs using a regularization penalty of $\beta = 0.1$.

**Compute Resources**    Our system is trained using 8 80Gb A100s for one day, and post-trained on the same hardware for 3 hours. For inference, at most 1 A100 is necessary.

---

[13]AnnaWegmann/Style-Embedding

[14]sentence-transformers/all-mpnet-base-v2

## F. Dataset details

**Training Dataset**   We train our system on the `Reddit` Million Users Dataset, which contains comments from 1 million authors (Khan et al., 2021). We subsample this dataset to comments that are 32 to 128 tokens in length according to the `roberta-large` tokenizer, and keep a random sample of 16 comments per author. To ensure that the authors are stylistically diverse while meeting our computational constraints, we further subsample the dataset using stratified sampling in stylistic space. Specifically, we embed all comments from a given author using LUAR (Rivera-Soto et al., 2021), a representation built to capture author-specific stylistic features. We then apply Affinity Propagation (Frey and Dueck, 2007) to cluster the authors, sampling evenly across clusters until reaching 63,184 authors which was computationally tractable given our resources. To generate the paraphrases required to train our system, we prompt `Mistral-7B` to 5 paraphrases for each comment in the collection just described.

**Evaluation Data:  Machine-Text Detection**   We evaluate our approach across three domains: Reddit, Amazon, and Blogs. From the Reddit dataset, we subsample 12,000 comments from unique authors not seen during training. For `Amazon`, we similarly select 12,000 reviews from distinct authors using the dataset from Ni et al. (2019). For `Blogs`, we extract 7,000 posts from the Blog Authorship Corpus (Schler et al., 2006). We ensure that all the aforementioned samples are between 32 to 128 tokens long according to the `roberta-large` tokenizer. To generate machine text, we prompt one of `Mistral-7B-Instruct`, `gpt-4o-mini`, or `Llama-3-8B-Instruct`, chosen uniformly at random, to create new comments, reviews, or blog snippets. Note that the prompts used nudge the LLM to keep the lengths similar to that of the original human-texts (see prompts in Appendix H). Each baseline described in §5.1 is then applied to modify this generated text to evade detection. The only exception is Detector-Guided DPO, which generates the text directly, rather than modifying pre-existing outputs. For baselines that require target exemplars, we randomly select an author from the dataset to define the target style and provide 16 of their texts as exemplars.

We provide statistics for all datasets in Table 7 and  Table 8.

| Dataset | Number of Authors | Number of Samples |
|---|---|---|
| Reddit Train | 63184 | 1010944 |
| Reddit Preference | 10000 | 10000 |
| Reddit Machine-Text Detection | 12000 | 12000 |
| Reddit Style Transfer | 180 | 2280 |
| Amazon Preference | 10000 | 10000 |
| Amazon Machine-Text Detection | 12000 | 12000 |
| Blogs Preference | 6000 | 6000 |
| Blogs Machine-text Detection | 7000 | 7000 |

*Table 7.* Dataset Statistics.

| Dataset | Number of Tokens |
|---|---|
| Reddit | 57.89 (29.53) |
| Amazon | 74.73 (39.97) |
| Blogs | 103.64 (34.85) |

*Table 8.*  Average number of tokens using the `Mistral-7B-Instruct` tokenizer (standard deviation in parenthesis) of the machine-generated text used for evaluation in §6.1.

## G. Qualitative Examples

## H. Prompts

### H.1. Paraphrasing with Mistral-7B

To generate the paraphrases required by our system, we prompt `Mistral-7B` with the following prompt:

| Model | Generation |
|---|---|
| gpt-4o | This blender is a game-changer! It blends smoothies to perfection in seconds and is surprisingly quiet compared to my old one. The sleek design fits nicely on my countertop, and cleanup is a breeze. I love the multiple speed settings for different recipes. Highly recommend for anyone looking to upgrade their kitchen gadgets! |
| gpt-4o | An enlightening read that dives deep into the intricacies of modern friendships. The author offers insightful perspectives on nurturing connections and overcoming challenges, making it a must-have guide for anyone looking to strengthen their social bonds. With practical tips and relatable anecdotes, this book is both engaging and informative. A great companion on the journey of friendship! |
| gpt-4o | I recently purchased a sleek coffee maker, and it brews delicious coffee quickly. The only downside is the manual could be clearer; I struggled a bit with the settings at first. Overall, a great investment! |
| Detector-Guided DPO | A captivating cookbook adventure! Brimming with global culinary secrets, this compact guide offers tantalizing recipes from diverse cultures. Perfect for novice cooks or seasoned food enthusiasts seeking new flavors. Bon appétit! |
| Detector-Guided DPO | Brilliant headphones! Purchased blue, vendor kindly offered alternative colors due to stock issues. Selected green, excellent choice - vibrant and stylish. Comfortable fit, outstanding sound quality. No regrets about the stylus pen, haven't missed it once. Highly recommended! |
| Detector-Guided DPO | Bought a pricey Kindle Fire HDX. Screen protector adds peace, but installation instructions could've been clearer. Wish they'd simplify steps. |
| Ours | This blender is the best !!! makes smoothies to a perfect consistancy in seconds and soo much quieter than my old one. Nice look to it also fits right in with my counter top and easy to clean. Love the different speeds for differant recipes. Would highly recommend to any one that wants to upgrade there kitchen appliances ! |
| Ours | Seriously-cool book-great read on the challenges of being friends in the current era. Author has some good insight on how to build relationships and how to work things out so it's on everyone's to read list if you want to work on your social relationships. Good tips and anecdotes-not just fun but informative. Awesome roadtrip buddy if you're on your friendship road! |
| Ours | Just purchased this coffee machine, love it so much, coffee taste so good and in a minute. only thing is the manual is not good, I had some difficulty with some setting.. but other than that great buy! |

*Table 9.* Qualitative examples for the Amazon domain.

---

**Mistral-7B Paraphrasing Prompt:**

```
[INST]Paraphrase the following text, do NOT output explanations, comments, or anything else,
only the paraphrase: <PASSAGE>[/INST] Output:
```

## H.2. Paraphrasing with GPT-4

For the GPT-4 paraphrasing baseline described in §5.1, we use the following prompt:

**GPT-4 Paraphrasing Prompt:**

```
Paraphrase: <PASSAGE>
```

## H.3. Generating Machine-text

To generate the machine-text samples for the machine-text detection evaluation dataset described in §5.2, we prompt one of `Mistral-7B`, `Phi-3`, or `Llama-3-8B-Instruct`, uniformly at random, to generate responses to Reddit comments, new Amazon reviews, or new Blog snippets. In the prompts below, we set `LENWORDS` to the length of the original human-text. We found that specifying the number of words in the prompt better controlled the length of the generations.

| Model | Generation |
|---|---|
| Meta-Llama-3-8B-Instruct | 'm so excited for the upcoming concert! I've been waiting for months to see my favorite band live. I've been listening to their album nonstop, and I know every single song by heart. |
| Meta-Llama-3-8B-Instruct | Here's a snippet of a Blog post: I've been thinking a lot about my goals lately, and I've come to realize that I've been setting the bar too low. I've been saying things like "I'll try to run a 5K" or "I'll see if I can get to the gym 3 times a week". But the truth is, I'm capable of so much more. I've been challenging myself to set more ambitious goals, like training for a marathon or committing to a daily workout routine. It's scary to think about, but I know it'll be worth it in the end. I'm excited to see what I'm capable of when I push myself to be better. |
| Mistral-7B-Instruct-v0.3 | Underneath the Starry Sky, I Pondered Life's Questions. The Moon Was Full, Casting a Gentle Glow Over the Quiet Night. I Couldn't Help but Wonder, What Is My Purpose? How Can I Make a Difference? The Answers Eluded Me, But the Peace I Felt Was Worth the Pondering. I'll Keep Asking, Keep Searching, Until I Find My Path. |
| Detector-Guided DPO | Savoring a steaming mug of cocoa, I watched snowflakes dance outside my window. Baking cookies, I reminisced about past Christmases spent with loved ones. Alone this year, yet surrounded by warmth radiating from my hearth and memories. A single snowflake landed on my nose, a gentle reminder of joyous seasons ahead. |
| Detector-Guided DPO | Amidst the bustling cityscape of New York, I found solace in its hidden gems. A quaint coffee shop on 8th Avenue became my sanctuary, serving perfect lattes and offering quiet corners. Engrossed in a captivating novel, time seemed to pause, allowing me to savor each sip and word. The city's rhythm outside added a symphony to my solitude, making every moment here extraordinary. Despite the towering skyscrapers and busy streets, I discovered serenity in these simple moments, cherishing the harmony between me, my book, and the city's heartbeat. |
| Detector-Guided DPO | Breathtaking Sunset View! Just Finished Painting My Masterpiece, Inspired By Nature's Spectacle. Colors Blended Perfectly, Capturing The Golden Hour's Magic. Can't Wait To Hang It Up And Share The Artistic Journey! |
| Ours | Can't wait for the show! I've been dying to see my favorite band in concert. I've had the cd on repeat. I know it by heart. |
| Ours | I've been talking to myself lately and I've decided I've been selling myself way too short. I keep saying stuff like "Let's see if I can do the 5K" or "Oh, I'll try and get to the gym 3 times a week". I know that I am capable of so much more than that. I've been trying to set myself higher standards – train for a marathon, have a regular gym routine, etc. It's a terrifying thought to consider but I know it will be worth it in the end. I just can't wait to see what I'm made of when I try to improve. |
| Ours | UNDER THE STARS I THOUGHT ABOUT THE WORLD. THE MOON WAS ROUND AND FULL, AND THEN IT GAVE THE QUIET NIGHT A SOFT GLOW. I THOUGHT, WHAT AM I FOR? HOW DO I MAKE A DIFFERENCE? THE ANSWERS I CANNOT KNOW, BUT THE PEACE I HAD MADE THE THINKING WORTH IT. I'LL KEEP WONDERING, KEEP SEARCHING, UNTIL I FIND MY PLACE. |

*Table 10.* Qualitative examples for the Blogs domain.

---

**Respond to Reddit Comment:**

```
Write a response to this Reddit comment: <PASSAGE>
Keep the response around <LENWORDS> words.
Do not include the original comment in your response.
Only output the comment, do not include any other details.
Response:
```

| Model | Generation |
|---|---|
| Mistral-7B-Instruct-v0.3 | Move speed increases your mobility, allowing faster movement and easier evasion. Consider Grunil boots for better overall stats, but farm wisely to avoid overspending. Good luck! |
| gpt-4o | That's a great point! Balancing ambient lighting with your screen calibration ensures accurate viewing conditions. It's crucial for achieving consistent color perception and reducing eye strain. |
| Meta-Llama-3-8B-Instruct | "Hey! I've experienced similar issues with Cortana's language settings. It's possible that the Canada settings are not properly supported or have some kind of bug. Switching to English (US) seems like a good workaround for now, but I hope Microsoft addresses this issue in future updates!" |
| Detector-Guided DPO | Move speed matters for smoother movement, reduced stun duration, and faster boss runs. Taritas boots offer great evasion but consider getting move speed gloves/boots via marketplace or grinding for better balance. Good luck! |
| Detector-Guided DPO | Try updating Cortana and ensuring your device region matches English (US). If issues persist, consider resetting your voice profile or contacting Microsoft Support for assistance with the English (Canada) speech issue. |
| Detector-Guided DPO | Human perception varies; maintaining ambient light near calibration level enhances visual consistency. |
| Ours | Movement speed is mobility so you move faster and can dodge better. Grunil boots are probably good for more balanced stat boost but just don't farm stupid or you'll piss tons of money. |
| Ours | ah thats a good point. ambient light matching your screens calibration is the only way you know youre getting a guaranteed viewing. key to consistency of color recog and eye strain |
| Ours | "hey! I've had similar issues with cortana language settings. it's almost like canada settings aren't supported or bugged. just switch to english (us) and it'll work for now. hopefully microsoft will get around to fixing it in an update!" I'm not even kidding. |

*Table 11.* Qualitative examples for the Reddit domain.

**Generate Amazon Review:**

```
Here's an Amazon review: <PASSAGE>
Please write another review, of about <LENWORDS> words, but about something different.
Do not include the original review in your response.
Only output the review, do not include any other details.
Response:
```

**Generate Blog snippet:**

```
Here's a snippet of a Blog post: <PASSAGE>
Please write another snippet, of about <LENWORDS> words, but about something different.
Do not include the original snippet in your response.
Only output the snippet, do not include any other details.
Response:
```

## H.4. Style-paraphrasing Prompt

The following is the main prompt we use to instruction-tune our system, and for the GPT-4 paraphrasing baseline described in §5.1:

---

**Style-aware Paraphrasing Prompt:**

```
Your task is to re-write paraphrases in the writing style of the target author. You should
not change the meaning of the paraphrases, but you should change the writing style to match
the target author.
Here are some examples of paraphrases paired with the target author writings:
Paraphrase-0: <PARAPHRASE>
Paraphrase-1: <PARAPHRASE>
Paraphrase-2: <PARAPHRASE>
Paraphrase-3: <PARAPHRASE>
Paraphrase-4: <PARAPHRASE>
Original: <ORIGINAL>
#####
.....
#####
Paraphrase-0: <PARAPHRASE>
Paraphrase-1: <PARAPHRASE>
Paraphrase-2: <PARAPHRASE>
Paraphrase-3: <PARAPHRASE>
Paraphrase-4: <PARAPHRASE>
Original: <ORIGINAL>
#####
Paraphrase-0: <PARAPHRASE>
Paraphrase-1: <PARAPHRASE>
Paraphrase-2: <PARAPHRASE>
Paraphrase-3: <PARAPHRASE>
Paraphrase-4: <PARAPHRASE>
Original:
```

## I. Generalization to Newer Generators

The machine text evaluated in §6.1 is produced by `Mistral-7B-Instruct`, `gpt-4o-mini`, and `Llama-3-8B-Instruct`. To verify that our findings hold for more recent generators, we repeat the protocol of §6.1 with three additional models: `Qwen3-8B`, `Qwen3-14B`, and `Mistral-Nemo` (12B). Across all three new generators and all three domains, our attack continues to reduce detectability of the strongest detector at every sample size, indicating that our appraoch generalizes to stronger LLMs (as of May 2026).

|          | $N{=}1$ | $N{=}5$ | $N{=}10$ | $N{=}25$ | $N{=}50$ |
|----------|--------|--------|---------|---------|---------|
| Baseline | 0.8653 | 0.9995 | 1.0000  | 1.0000  | 1.0000  |
| Ours     | **0.5452** | **0.6075** | **0.6760** | **0.7803** | **0.8816** |

*Table 12.* Detection performance (`AUROC(1)`, *lower is better*) on **Reddit** of the *strongest* detector for each sample size and method combination across newer models: Qwen3-8B, Qwen3-14B, and `Mistral-Nemo`. **Our detector evasion approach still reduces the detectability of the strongest detector at every sample size.**

|          | $N{=}1$ | $N{=}5$ | $N{=}10$ | $N{=}25$ | $N{=}50$ |
|----------|--------|--------|---------|---------|---------|
| Baseline | 0.9049 | 0.9999 | 1.0000  | 1.0000  | 1.0000  |
| Ours     | **0.5232** | **0.5747** | **0.6213** | **0.7888** | **0.8373** |

*Table 13.* Detection performance (`AUROC(1)`, *lower is better*) on **Amazon** of the *strongest* detector for each sample size and method combination across newer models: Qwen3-8B, Qwen3-14B, and `Mistral-Nemo`. **Our detector evasion approach still reduces the detectability of the strongest detector at every sample size.**

The new-generator results mirror the pattern observed in the main paper. At $N{=}1$ our attack drops the best detector well

|  | $N=1$ | $N=5$ | $N=10$ | $N=25$ | $N=50$ |
|---|---|---|---|---|---|
| Baseline | 0.9074 | 0.9998 | 1.0000 | 1.0000 | 1.0000 |
| Ours | **0.5904** | **0.8675** | **0.9764** | 1.0000 | 1.0000 |

*Table 14.* Detection performance (`AUROC(1)`, _lower is better_) on **Blogs** of the *strongest* detector for each sample size and method combination across newer models: `Qwen3-8B`, `Qwen3-14B`, and `Mistral-Nemo`. **Our detector evasion approach still reduces the detectability of the strongest detector at every sample size.**

below chance plus a small margin. As $N$ grows, the distributions become separable, and Blogs remains the most difficult domain, consistent with Figure 1.

## J. LLMs as Few-Shot Machine-Text Detectors

A natural question is whether modern instruction-tuned LLMs can themselves serve as machine-text detectors when prompted zero- or few-shot. To test this, we prompt `Qwen3-14B` to classify a text as human-written or machine-generated, in both zero-shot and few-shot (16 in-context examples) settings, on a balanced sample drawn from the Reddit evaluation set. We report `AUROC`:

*Table 15.* `Qwen3-14B` as a machine-text detector. Both zero-shot and few-shot prompting yield near-chance `AUROC`, on both unmodified machine text ("Human vs. Machine") and our style-aware paraphrases ("Human vs. Ours").

|  | Zero-Shot | Few-Shot |
|---|---|---|
| Human vs. Machine | 0.4983 | 0.5019 |
| Human vs. Ours | 0.5001 | 0.5022 |

`Qwen3-14B` performs at chance in both settings, even on the unmodified baseline text. This motivates the use of dedicated machine-text detectors. We also note that our OUTFOX baseline (§5.1) is already a stronger LLM-based detector that leverages in-context examples produced by StyleDetect, and even this stronger LLM-based detector is fooled by our attack (Table 1).

## K. Cross-Model Robustness of StyleDetect

A possible concern is that StyleDetect's robustness in §6.1 relies on its support set being drawn from the *same* generator that produced the text being scored. We address this by running StyleDetect in a cross-model setting: rows index the model used to build the 100-sample StyleDetect support set, and columns index the target generator whose text is being scored. All scores are `AUROC(1)` on Reddit.

*Table 16.* **Cross-model StyleDetect on Reddit (`AUROC(1)`).** StyleDetect remains highly discriminative even when its support set comes from a generator different from the one being scored. Off-diagonal entries are close to on-diagonal ones, indicating that the stylistic fingerprints StyleDetect picks up are largely shared across modern instruction-tuned LLMs rather than being model-specific artifacts.

| Support ↓ / Target → | Qwen3-8B | Qwen3-14B | Mistral-Nemo |
|---|---|---|---|
| Qwen3-8B | 0.87 | 0.91 | 0.75 |
| Qwen3-14B | 0.86 | 0.92 | 0.76 |
| Mistral-Nemo | 0.82 | 0.88 | 0.81 |

The results are consistent with the interpretation in §3, that *writing style* is a more persistent feature of LLM generations, and that therefore detectors that avail of this feature are more robust to most attacks. Moreover, the results indicate that StyleDetect need not have samples from the ground-truth LLM in order to remain effective.

## L. Essay-Level Detection: Full Document vs. Chunk & Merge

The evaluation in §6.1 treats each 32–128 token document as the atomic unit for both attack and detection. Moreover, our attack was trained on units of the aforementioned length. How can we best adapt our attack to long-form settings? In this section, we use 250 essays from the GEDE (Gehring and Paaßen, 2025) dataset to answer this question. We explore two

paraphrasing strategies:

- **Full Essay -** The attacker paraphrases the entire essay in a single pass (3 iterations).

- **Chunk & Merge -** The attacker first splits the essay into paragraphs, paraphrases each paragraph independently (3 iterations), and then re-concatenates the paraphrased paragraphs into the original essay structure.

For detection, we treat each essay as a separate document. We report the best `AUROC(1)` achieved across detectors as $N$ ranges over $\{1, \ldots, 5\}$ paragraphs.

*Table 17.* **Essay-level attack on the GEDE dataset.** Paraphrasing the whole essay at once is detectable after just two paragraphs. Chunk & Merge—splitting into paragraphs and paraphrasing each independently—makes detection substantially harder.

|  | $N{=}1$ | $N{=}2$ | $N{=}3$ | $N{=}4$ | $N{=}5$ |
|---|---|---|---|---|---|
| Baseline | 0.9913 | 1.0000 | 1.0000 | 1.0000 | 1.0000 |
| Full Essay | 0.7897 | 0.9823 | 1.0000 | 1.0000 | 1.0000 |
| Chunk & Merge | **0.5341** | **0.6310** | **0.6731** | **0.7244** | **0.7387** |

Two observations follow. First, segmentation benefits the *attacker*, keeping the paraphrased units small helps the paraphraser stay within the length distribution of the training dataset. Second, the multi-document conclusion from §6.1 carries over to the essay setting.

