# OpenReview forum: "Attacks on Machine-Text Detectors Retain Stylistic Fingerprints"
_ICML.cc/2026/Conference — ICML 2026 regular_

### Official Review · Reviewer_1YKP · 2026-03-10

**Soundness:** 3
**Presentation:** 3
**Significance:** 2
**Originality:** 3
**Overall Recommendation:** 5
**Confidence:** 3

**Summary:**

This paper investigates the robustness of style features in machine-generated text detection and the limits of detection evasion strategies. The authors first demonstrate that while existing attack methods can reduce the performance of standard detectors, they cannot eliminate the underlying style "fingerprint" of machine text. To address this issue, the authors propose a novel style-aware paraphraser. This model undergoes two-stage training using supervised fine-tuning (SFT) and direct preference optimization (DPO), enabling it to mimic the writing style of specific human authors while preserving the original semantics, thereby evading detection. Experiments show that this method successfully evades all tested detectors at the single-document level.

**Compliance With Llm Reviewing Policy:**

Affirmed.

**Final Justification:**

I have read the authors’ rebuttal and the clarifications, and appreciate their great efforts. As my concerns have been well addressed, I will update my current score accordingly.

**Key Questions For Authors:**

N.A.

**Limitations:**

Yes.

**Strengths And Weaknesses:**

Strength:

1.	The paper reveals that general optimization and conventional paraphrasing attacks cannot erase the style fingerprint of a model, and proves the robustness of the style feature space in defending against current mainstream evasion attacks.
2.	The proposed method achieving state-of-the-art (SOTA) evasion performance under single-sample detection mechanisms while maintaining high semantic fidelity.
Weakness:
1.	Highly dependent on the target author's reference text: This attack method requires obtaining sample texts from human authors to demonstrate diverse writing styles. In many real-world adversarial scenarios, attackers often struggle to obtain sufficient and high-quality corpora from a specific target author.
2.	In the evaluation protocol, StyleDetect used samples from a large base model of 100 targets as its support set, while zero-shot detectors such as FastDetectGPT and Binoculars completely lack prior information about the target distribution. Although the authors stated that this setting was intended to isolate and test "style fingerprint" features, this experimental design actually gave StyleDetect a significant informational advantage, potentially exaggerating the absolute performance of this few-shot detector relative to zero-shot detectors in cross-sectional comparisons.
3.	The paper mentions in its limitations that LLM-generated paraphrasing introduces semantic drift. Because this method requires iteratively applying the paraphrasing model to gradually reduce the distributional differences between machine and human text, the paper lacks a specific quantitative analysis of the accumulation of factual errors or deep semantic loss during this iterative process with increasing iterations.
4.	The authors evaluated several open- and closed-source models, e.g., Qwen-2.5-7B-Instruct and gpt-4o-mini. However, given the rapid development of large language models, these specific versions are now outdated. I recommend that the authors incorporate the latest generation of models, such as the GPT-5 and Qwen 3 series, into their experimental evaluation.

---

> ### Author Rebuttal · Authors · 2026-03-31
>
> We thank R4 for recognizing the SOTA evasion performance and the novelty of the two-stage SFT+DPO approach.
>
> > Highly dependent on the target author's reference text [...] attackers often struggle to obtain sufficient and high-quality corpora from a specific target author.
>
> We note that in many domains, obtaining such text is straightforward.
> For example, in social media domains such as Reddit, Twitter, and blogs, the target author's writing is publicly available.
> Even in news media or academic settings (ArXiv), collecting 10 or more documents from a given author is typically feasible.
> There is no special requirement for the target author's writing samples, indeed, we chose random authors from each domain to style transfer to.
>
> > StyleDetect used samples from a large base model of 100 targets as its support set [...] this experimental design actually gave StyleDetect a significant informational advantage, potentially exaggerating the absolute performance.
>
> We address this in Section 3, where we introduce SemDetect as a control: it uses the *same* 100 support examples but relies on semantic rather than stylistic representations.
> We find that SemDetect has lower performance than StyleDetect, confirming that the robustness stems from stylistic features specifically, not from the informational advantage of the few-shot setup.
>
> > The paper lacks a specific quantitative analysis of the accumulation of factual errors or deep semantic loss during this iterative process with increasing iterations.
>
> Good point.
> We have measured the semantic similarity between the generated text and the original at each iteration of the attack:
>
> ### Reddit: Similarity to Original Machine Text
>
> | Stage | SBERT Cosine Sim |
> |------|-----------------|
> | Paraphrase (input) | 0.90 (0.09) |
> | Iteration 1 | 0.9330 (0.0838) |
> | Iteration 2 | 0.8985 (0.0991) |
> | Iteration 3 | 0.8836 (0.1077) |
>
> ### Amazon: Similarity to Original Machine Text
>
> | Stage | SBERT Cosine Sim |
> |------|-----------------|
> | Paraphrase (input) | 0.96 (0.04) |
> | Iteration 1 | 0.9602 (0.0364) |
> | Iteration 2 | 0.9330 (0.0522) |
> | Iteration 3 | 0.9206 (0.0643) |
>
> ### Blogs: Similarity to Original Machine Text
>
> | Stage | SBERT Cosine Sim |
> |------|-----------------|
> | Paraphrase (input) | 0.92 (0.1) |
> | Iteration 1 | 0.9331 (0.1064) |
> | Iteration 2 | 0.8953 (0.1131) |
> | Iteration 3 | 0.8767 (0.1153) |
>
> Across all three domains, semantic similarity remains high (above 0.87) even after three iterations, indicating that the iterative process does not introduce substantial semantic drift.
>
> We will add these results to the Appendix of the camera-ready version.
>
> > The authors evaluated several open- and closed-source models [...] these specific versions are now outdated. I recommend that the authors incorporate the latest generation of models, such as the GPT-5 and Qwen 3 series.
>
> We have added experiments with newer generators (Qwen3-8B, Qwen3-14B, and Mistral-Nemo).
> Our attack remains effective at humanizing text from these more recent models:
>
>
> ### New Models Average (Qwen3-8B, Qwen3-14B, Mistral-Nemo) — Reddit (AUROC @ 1% FPR)
> |          |      1 |      5 |     10 |     25 |     50 |
> |:---------|-------:|-------:|-------:|-------:|-------:|
> | Baseline | 0.8653 | 0.9995 | 1.0000 | 1.0000 | 1.0000 |
> | Ours     | 0.5452 | 0.6075 | 0.6760 | 0.7803 | 0.8816 |
>
>
> ### New Models Average (Qwen3-8B, Qwen3-14B, Mistral-Nemo) — Amazon (AUROC @ 1% FPR)
> |          |      1 |      5 |     10 |     25 |     50 |
> |:---------|-------:|-------:|-------:|-------:|-------:|
> | Baseline | 0.9049 | 0.9999 | 1.0000 | 1.0000 | 1.0000 |
> | Ours     | 0.5232 | 0.5747 | 0.6213 | 0.7888 | 0.8373 |
>
>
> ### New Models Average (Qwen3-8B, Qwen3-14B, Mistral-Nemo) — Blogs (AUROC @ 1% FPR)
> |          |      1 |      5 |     10 |     25 |     50 |
> |:---------|-------:|-------:|-------:|-------:|-------:|
> | Baseline | 0.9074 | 0.9998 | 1.0000 | 1.0000 | 1.0000 |
> | Ours     | 0.5904 | 0.8675 | 0.9764 | 1.0000 | 1.0000 |

---

> > ### Author Rebuttal · Reviewer_1YKP · 2026-04-06
> >
> > I have read the authors’ rebuttal and the clarifications, and appreciate their great efforts. As my concerns have been well addressed, I will update my current score accordingly.

---

### Official Review · Reviewer_h9om · 2026-03-11

**Soundness:** 3
**Presentation:** 3
**Significance:** 2
**Originality:** 2
**Overall Recommendation:** 4
**Confidence:** 4

**Summary:**

This paper focuses on the red-teaming of Machine-Text Detectors. It argues that existing attacks cannot erase the underlying stylistic “fingerprints" of machine text, which perform badly against few-shot detectors. The author proposes a style-aware paraphrasing attack that rewrites machine-text toward the style of a specific human using exemplar conditioning. They use DPO to optimize and do iterative paraphrasing at inference. Experiments show that their method evades all detectors in single-document settings, but performs worse if the number of documents increases.

**Compliance With Llm Reviewing Policy:**

Affirmed.

**Final Justification:**

After rebuttal, most of my concerns, except for W2, are resolved. I have raised my score to 4.

However, I still believe that multi-document detection is inherently asymmetric in favor of the attacker, which limits its practical applicability.

**Key Questions For Authors:**

- If a detector splits a long essay into multiple segments and treats them as separate documents, how does the detection performance compare to treating the essay as a single document?
- A similar question applies to the attack: how does the attack perform if a long essay is split into multiple segments before paraphrasing?

**Limitations:**

yes

**Strengths And Weaknesses:**

### Strength

- The finding that multi-document detection is more effective is interesting
- The paper is well written and easy to follow.
- The experiments are extensive, including quality and ablations.

### Weakness

- The proposed method lacks novelty. The approach of few-shot style-transfer is very similar to TinyStyler. The detector-oriented DPO is also a trivial approach for optimization attacks. The simple combination of exisiting method make the novelty incremental.
- Though promising, multi-document detection is not practical in real life. In most cases, the detector cannot get 10+ documents from the same person, which weakens the practical significance of this finding.
- Some deeper analysis would be helpful, such as examining how many documents are required for relatively robust detection or how text length affects the detection
- The method uses iterative paraphrasing with multiple candidates and semantic reranking, which may introduce significant inference overhead.

---

> ### Author Rebuttal · Authors · 2026-03-31
>
> > The proposed method lacks novelty. [...] The simple combination of existing methods make the novelty incremental.
>
> The **primary contribution** is not the method alone, but the *finding* that the stylistic feature space is fundamentally more robust than surface-level features, and that reliable detection requires multi-document analysis.
>
> Although our method is conceptually related to TinyStyler, (1) TinyStyler is consistently detected as machine-generated, and (2) in Appendix D our method outperforms TinyStyler in both stylistic similarity and semantic preservation.
>
> > Though promising, multi-document detection is not practical in real life. In most cases, the detector cannot get 10+ documents from the same person [...]
>
> Multi-document detection is already the norm in many settings: academic integrity involves portfolios of assignments, and content moderation platforms have post histories.
> The N=5-10 range is realistic in these scenarios.
> Moreover, the *attack* itself requires the target author's writing samples, so defender and attacker have symmetric information requirements.
>
> More broadly, we characterize *what* robust detection requires.
> If reliable detection matters, and in domains like academic integrity and journalism it clearly does, the detection paradigm must adapt.
> Our work identifies the direction.
>
> > Some deeper analysis would be helpful, such as examining how many documents are required for relatively robust detection or how text length affects the detection.
>
> Figure 1 already shows how detection varies with N.
> We add an experiment grouping texts into length tertiles per domain and computing AUROC(1) for each.
>
> ### Reddit — Short (≤30w)
> |                          |    1 |    5 |   10 |   25 |
> |:-------------------------|-----:|-----:|-----:|-----:|
> | Baseline                 | 0.81 | 1.00 | 1.00 | 1.00 |
> | Detector-Guided DPO      | 0.88 | 1.00 | 1.00 | 1.00 |
> | OUTFOX                   | 0.92 | 1.00 | 1.00 | 1.00 |
> | Paraphrasing             | 0.85 | 1.00 | 1.00 | 1.00 |
> | DIPPER                   | 0.62 | 0.99 | 1.00 | 1.00 |
> | Prompting (Style-aware)  | 0.80 | 1.00 | 1.00 | 1.00 |
> | TinyStyler               | 0.68 | 0.99 | 1.00 | 1.00 |
> | Ours | 0.53 | 0.61 | 0.64 | 0.80 |
>
>
> ### Reddit — Medium (31–46w)
>
> |                          |    1 |    5 |   10 |   25 |
> |:-------------------------|-----:|-----:|-----:|-----:|
> | Baseline                 | 0.95 | 1.00 | 1.00 | 1.00 |
> | Detector-Guided DPO      | 0.98 | 1.00 | 1.00 | 1.00 |
> | OUTFOX                   | 0.99 | 1.00 | 1.00 | 1.00 |
> | Paraphrasing             | 0.98 | 1.00 | 1.00 | 1.00 |
> | DIPPER                   | 0.61 | 0.97 | 1.00 | 1.00 |
> | Prompting (Style-aware)  | 0.95 | 1.00 | 1.00 | 1.00 |
> | TinyStyler               | 0.66 | 0.97 | 1.00 | 1.00 |
> | Ours | 0.53 | 0.56 | 0.59 | 0.74 |
>
>
> ### Reddit — Long (>46w)
> |                          |    1 |    5 |   10 |   25 |
> |:-------------------------|-----:|-----:|-----:|-----:|
> | Baseline                 | 0.94 | 1.00 | 1.00 | 1.00 |
> | Detector-Guided DPO      | 1.00 | 1.00 | 1.00 | 1.00 |
> | OUTFOX                   | 1.00 | 1.00 | 1.00 | 1.00 |
> | Paraphrasing             | 0.99 | 1.00 | 1.00 | 1.00 |
> | DIPPER                   | 0.60 | 0.99 | 1.00 | 1.00 |
> | Prompting (Style-aware)  | 0.96 | 1.00 | 1.00 | 1.00 |
> | TinyStyler               | 0.68 | 0.97 | 1.00 | 1.00 |
> | Ours | 0.54 | 0.62 | 0.71 | 0.95 |
>
> Performance increases with N and length; most methods are more detectable with longer texts even at N=1. Similar trends hold for Amazon and Blogs.
>
> > The method uses iterative paraphrasing with multiple candidates and semantic reranking, which may introduce significant inference overhead.
>
> We agree. From the defense perspective, this overhead is beneficial: it raises the cost for attackers, discouraging deployment at scale.
>
> > Q: If a detector splits a long essay into multiple segments and treats them as separate documents, how does the detection performance compare to treating the essay as a single document? A similar question applies to the attack.
>
> We collected 250 essays from the GEDE[1] dataset and compared two strategies: (1) **Full Essay** (paraphrase the whole document) and (2) **Chunk & Merge** (split into paragraphs, paraphrase each independently, re-merge).
> Both use 3 iterations. We report best AUROC(1) across detectors:
>
> |               |      1 |      2 |      3 |      4 |      5 |
> |:--------------|-------:|-------:|-------:|-------:|-------:|
> | Baseline      | 0.9913 | 1.0000 | 1.0000 | 1.0000 | 1.0000 |
> | Chunk & Merge | 0.5341 | 0.6310 | 0.6731 | 0.7244 | 0.7387 |
> | Full Essay    | 0.7897 | 0.9823 | 1.0000 | 1.0000 | 1.0000 |
>
> Segmentation benefits the *attacker*: full-essay paraphrasing is detectable at N=2 (AUROC 0.98), while chunking makes detection substantially harder (0.53 at N=1, 0.73 at N=5).
>
> [1] Gehring, L., & Paaßen, B. (2025). Assessing LLM Text Detection in Educational Contexts: Does Human Contribution Affect Detection? arXiv preprint arXiv:2508.08096

---

> > ### Author Rebuttal · Reviewer_h9om · 2026-04-04
> >
> > Thanks for the authors for the detailed rebuttal and extra experiments. W1, W3 are solved.Q1 presents an interesting finding, and I encourage the authors to include it in paper.
> >
> > However, I remain unconvinced about W2. The claim that “the defender and attacker have symmetric information requirements” does not imply they can achieve the same information in practice. Appearantly, the attacker is in an active position—they can choose and control what information to use and provide. The defender does not have this flexibility. For example, a student submitting an essay can leverage many personal documents during the attack, while only exposing limited samples to the detector. This asymmetry raises concerns about the practical feasibility of the approach.
> >
> > For W4, the actual computational overhead should be reported.
> >
> > Thus, I maintian my current score.
> >
> > ---
> >
> > Thanks for the author's explaination. I remain unconvinced about W2. For instance, the student essay, the student can always access substantially more personal documents than the limited set of submissions available to the instructor, creating a fundamental asymmetry that undermines multi-document detection.
> >
> > W4 is solved, and I encourage the authors to include it in the paper.
> >
> > As most of my concerns, except for W2, are resolved. I have raised my score accordingly.

---

> > > ### Author Response · Authors · 2026-04-05
> > >
> > > We thank Reviewer h9om for the follow-up and for acknowledging that W1 and W3 are resolved.
> > > We will add the results of the Q1 experiments to the camera-ready version, as suggested by the reviewer.
> > >
> > > **W2 (Practicality of multi-document detection):**
> > >
> > > We appreciate the reviewer's point about attacker/defender asymmetry, but we respectfully disagree that this undermines the contribution.
> > >
> > > Our paper makes three primary contributions: (1) the finding that the stylistic feature space is robust against existing attacks, (2) a method for generating text that evades even strong stylistic detectors, and (3) the finding that robust detection against such attacks requires multi-document analysis.
> > > We *characterize what robust detection requires*; we do not claim it is trivially deployable in every scenario.
> > > Identifying the conditions under which detection is reliable, and those under which it fails, is a valuable contribution regardless of whether every deployment setting satisfies those conditions.
> > >
> > > That said, we note that the reviewer's example of a student essay actually describes a setting where multi-document detection is feasible.
> > > We elaborate on several concrete scenarios below:
> > > - **Academic integrity:** Instructors collect multiple assignments from each student over the course of a semester. A university could retroactively analyze all the student's submissions (essays, problem sets, discussion posts, etc.), yielding multiple documents with no additional collection effort.
> > > - **Peer review:** Peer-review systems have reviewers submit multiple reviews for the same conference or journal. Program chairs or ethics committees investigating suspicious reviews could pool multiple reviews per reviewer from a single venue.
> > > - **Social media and content moderation:** Platforms such as Reddit and Twitter retain full post histories. Therefore, detectors have access to multiple posts per user. Our Reddit, Amazon reviews, and Blog experiments directly model this setting.
> > >
> > > In each of these cases, multi-document analysis requires no special data collection, the documents already exist as a byproduct of normal activity.
> > > Thus, in many real-world settings, multi-document analysis is feasible.
> > > By shifting from a single-document prediction to an *author-level* prediction that aggregates multiple documents, our findings show that more reliable detection is possible.
> > >
> > > **W4 Timing of Methods**
> > >
> > > We report per-sample inference times measured with a batch size of 1 on a single NVIDIA A100 (80GB) GPU, averaged over 100 samples after a 5-sample warmup.
> > > We exclude API-based methods (OUTFOX, Style-Aware Prompting, and Paraphrasing via gpt-4o-mini) since their latency reflects server-side infrastructure rather than the computational cost of the method itself.
> > >
> > > | Method | s/sample | Notes |
> > > |--------|----------|-------|
> > > | Detector-Guided DPO (Nicks et al.) | 0.55 | vLLM, Mistral-7B |
> > > | DIPPER | 1.25 | HuggingFace Transformers, T5-XXL |
> > > | TinyStyler | 1.53 | HuggingFace Transformers, T5 |
> > > | Ours (total, 3 iterations) | 5.64 | vLLM, 3 iterations (generation + SBERT re-ranking), 10 candidates per iteration, Mistral-7B |
> > >
> > > We note that DIPPER and TinyStyler use HuggingFace Transformers for inference, as they're not supported by dedicated inference engines such as vLLM.
> > > Therefore, they do not benefit from the optimizations available in vLLM.
> > > In contrast, both our method and the Detector-Guided DPO baseline use vLLM.
> > >
> > > We will add these timings to the camera-ready version.

---

### Official Review · Reviewer_wC2Q · 2026-03-12

**Soundness:** 3
**Presentation:** 4
**Significance:** 3
**Originality:** 3
**Overall Recommendation:** 4
**Confidence:** 4

**Summary:**

The paper is mainly studying for the continuous game between machine text inspection and evasion attacks. The core point of the paper is that while common attacks can significantly weaken the metrics of detection methods, they cannot erase the "style fingerprint" of machine-generated text. Therefore, we first train a paraphraser that conditionalizes the style of the target author using the "human-machine rewriting" pair, and then use a specially trained detector to construct the preference data for DPO. At inference time, we also use iterative candidate generation and semantic filtering to narrow the training/test distribution difference.

**Compliance With Llm Reviewing Policy:**

Affirmed.

**Final Justification:**

I appreciate the authors’ thoughtful rebuttal and the additional clarifications. The response improves my understanding of the paper, especially regarding the explanation of the performance drop under 90° rotation and the clarification that the runtime comparison was conducted under the same hardware setting. However, my concerns are only partially resolved at this stage. In particular, the explanation for the rotation vulnerability is plausible, but it is still not directly validated by additional evidence in the current manuscript. In addition, while the runtime fairness issue is clarified in the rebuttal, this experimental setup is not yet clearly documented in the paper itself. The discussion of recent developments in diffusion watermarking also remains somewhat limited. Overall, I appreciate the authors’ effort, but since these issues are not yet fully addressed in the manuscript, I will maintain my current score accordingly.

**Key Questions For Authors:**

1.The paper highlights multi-document detection as a future direction. Then is the minimum number of documents N stable for recovering detectability across different domain/author/text lengths?
2.Does the core conclusion about "style features are more robust" still hold when given no StyleDetect target model support examples, or only cross-model support examples? Does the current setup overestimate the advantages of style-based detectors?

**Limitations:**

Yes

**Strengths And Weaknesses:**

**Srtength**:
1.The motivation of the paper is clear, the combination of style conditional paraphrasing, DPO for its own output detectability and iterative reasoning is self-consistent, and explains why the authors' approach is better than just optimizing preference for zero-shot detectors.
2.The article is well presented, the graph serves the argument rather than the results, and the reader can clearly understand what the author is answering.
3.The entry point of this article has insight, and many avoidance methods only operate on surface statistical features, without touching the author's style space. Therefore, the detection research should change from single-document analysis to multi-document analysis.

**Weakness**:
1.Although the starting point has insight, the method itself is more of an engineering combination.
2.It is an important conclusion that multiple documents can help detection, but the analysis is not systematic enough, and the change of N of multiple documents under different domains, different authors, different lengths, and different target style richness is not clear for the time being.

---

> ### Author Rebuttal · Authors · 2026-03-31
>
> > Although the starting point has insight, the method itself is more of an engineering combination.
>
> We address novelty in our response to R3.
> In brief: (1) the stylistic feature space is fundamentally more robust than surface-level features, (2) ours is the first method to *jointly* optimize for style mimicry and undetectability, and (3) multi-document analysis recovers detectability.
> Moreover, our finding that the stylistic feature space is robust to optimization via DPO (Section 3) disproves the findings of [1].
>
> [1] Charlotte Nicks, et al. 2024. Language model detectors are easily
> optimized against. ICLR.
>
> > [...] the analysis is not systematic enough, and the change of N of multiple documents under different domains, different authors, different lengths, and different target style richness is not clear for the time being.
>
> Note that Figure 1 already shows how detection performance changes as N increases, broken down by domain.
> To further address this concern, we grouped texts into tertiles by length and compute AUROC(1) for each tertile:
>
> ### Reddit — Short (≤30w)
>
> |                          |    1 |    5 |   10 |   25 |   50 |
> |:-------------------------|-----:|-----:|-----:|-----:|-----:|
> | Baseline                 | 0.81 | 1.00 | 1.00 | 1.00 | 1.00 |
> | Detector-Guided DPO      | 0.88 | 1.00 | 1.00 | 1.00 | 1.00 |
> | OUTFOX                   | 0.92 | 1.00 | 1.00 | 1.00 | 1.00 |
> | Paraphrasing             | 0.85 | 1.00 | 1.00 | 1.00 | 1.00 |
> | DIPPER                   | 0.62 | 0.99 | 1.00 | 1.00 | 1.00 |
> | Prompting (Style-aware)  | 0.80 | 1.00 | 1.00 | 1.00 | 1.00 |
> | TinyStyler               | 0.68 | 0.99 | 1.00 | 1.00 | 1.00 |
> | Ours | 0.53 | 0.61 | 0.64 | 0.80 | 0.93 |
>
> ### Reddit — Medium (31–46w)
>
> |                          |    1 |    5 |   10 |   25 |   50 |
> |:-------------------------|-----:|-----:|-----:|-----:|-----:|
> | Baseline                 | 0.95 | 1.00 | 1.00 | 1.00 | 1.00 |
> | Detector-Guided DPO      | 0.98 | 1.00 | 1.00 | 1.00 | 1.00 |
> | OUTFOX                   | 0.99 | 1.00 | 1.00 | 1.00 | 1.00 |
> | Paraphrasing             | 0.98 | 1.00 | 1.00 | 1.00 | 1.00 |
> | DIPPER                   | 0.61 | 0.97 | 1.00 | 1.00 | 1.00 |
> | Prompting (Style-aware)  | 0.95 | 1.00 | 1.00 | 1.00 | 1.00 |
> | TinyStyler               | 0.66 | 0.97 | 1.00 | 1.00 | 1.00 |
> | Ours                     | 0.53 | 0.56 | 0.59 | 0.74 | 0.98 |
>
> ### Reddit — Long (>46w)
>
> |                          |    1 |    5 |   10 |   25 |   50 |
> |:-------------------------|-----:|-----:|-----:|-----:|-----:|
> | Baseline                 | 0.94 | 1.00 | 1.00 | 1.00 | 1.00 |
> | Detector-Guided DPO      | 1.00 | 1.00 | 1.00 | 1.00 | 1.00 |
> | OUTFOX                   | 1.00 | 1.00 | 1.00 | 1.00 | 1.00 |
> | Paraphrasing             | 0.99 | 1.00 | 1.00 | 1.00 | 1.00 |
> | DIPPER                   | 0.60 | 0.99 | 1.00 | 1.00 | 1.00 |
> | Prompting (Style-aware)  | 0.96 | 1.00 | 1.00 | 1.00 | 1.00 |
> | TinyStyler               | 0.68 | 0.97 | 1.00 | 1.00 | 1.00 |
> | Ours | 0.54 | 0.62 | 0.71 | 0.95 | 1.00 |
>
> Performance increases with both N and document length, note how most of the methods become more detectable with longer documents even at N=1.
> Similar trends hold for Amazon and Blogs.
>
> > Q1: Is the minimum number of documents N stable for recovering detectability across different domain/author/text lengths?
>
> See above.
> In practice, given that validation data is often available, we do not see this as a limitation, as it would be feasible to calibrate N for a given domain.
>
> > Q2: Does the core conclusion about "style features are more robust" still hold when given no StyleDetect target model support examples, or only cross-model support examples? Does the current setup overestimate the advantages of style-based detectors?
>
> In both Section 3 (lines 131-148) and Section 5.3 (lines 239-243), the support examples come from the *original, unoptimized* base LLM, not from attacked outputs.
> StyleDetect has no information about the attack.
> If the attacks erased the stylistic fingerprint, similarity to these anchors would drop and detection would fail; it does not.
> Moreover, SemDetect (Section 3, line 147) serves as a control using the same 100 support examples but semantic instead of stylistic representations.
> Its lower performance than StyleDetect shows that robustness comes from stylistic features, not the few-shot setup.
>
> We also ran StyleDetect cross-model (rows = support model, columns = target generator).
> It remains robust even cases when the support doesn't match the target (off-diagonals):
>
> ### AUROC@FPR=1% for StyleDetect
> | Support \ Target | Qwen3-8B | Qwen3-14B | Mistral-Nemo |
> |------------------|----------|-----------|---------------|
> | Qwen3-8B         | 0.87     | 0.91      | 0.75          |
> | Qwen3-14B        | 0.86     | 0.92      | 0.76          |
> | Mistral-Nemo     | 0.82     | 0.88      | 0.81          |

---

> > ### Author Rebuttal · Reviewer_wC2Q · 2026-04-03
> >
> > I appreciate the authors’ thoughtful rebuttal and the additional analyses. The response improves my understanding of the paper, especially by clarifying the role of cross-model support examples and by providing further results on how detection performance changes with the number of documents and document length. However, two key concerns remain only partially resolved in the current manuscript: first, the analysis of multi-document detection is still not fully systematic, as the minimum stable number of documents across different domains, authors, and style richness settings remains unclear; second, while the rebuttal provides evidence that style-based detection can remain effective beyond the same-model setting, it is still not fully established how strong the main conclusion is when target-model-specific support is unavailable. Overall, I appreciate the authors’ effort, but since these issues are not yet fully addressed in the paper itself, I will maintain my current score accordingly.

---

### Official Review · Reviewer_GGRk · 2026-03-13

**Soundness:** 3
**Presentation:** 3
**Significance:** 2
**Originality:** 3
**Overall Recommendation:** 4
**Confidence:** 4

**Summary:**

The paper explores methods for evading machine-generated text detectors by paraphrasing to emulate human style. They find that while this succeeds in single shot setting, it does not work when multiple examples are presented. They train an LM to generate human-like text and use multiple detectors to test the efficacy of the perturbations.

**Compliance With Llm Reviewing Policy:**

Affirmed.

**Final Justification:**

Many of the concerns are resolved, and I improved the originality score. My overall recommendation stays the same, as I am unsure how the method would scale to larger models and different families.

**Key Questions For Authors:**

1 - There are some studies that explored the direct use of LLMs to differentiate between machine code and human-generated code with signs of success [1,2,3,4]. Can the authors test their proposed stylistic change and the discovery about few-shot examples with these as baselines?

2 - How does the detection change when you use larger models in your StyleDetect pipeline?

3 - Any thoughts about quality after paraphrasing? The Table 8 example has one ‘Ours’ example that is in all caps; is this mimicking a specific human-author style?

[1] Panickssery, Arjun, Samuel Bowman, and Shi Feng. "Llm evaluators recognize and favor their own generations." Advances in Neural Information Processing Systems 37 (2024): 68772-68802.

[2] Ackerman, Christopher, and Nina Panickssery. "Inspection and control of self-generated-text recognition ability in llama3-8b-instruct." arXiv preprint arXiv:2410.02064 (2024).

[3] Mahbub, Taslim, and Shi Feng. "Mitigating Self-Preference by Authorship Obfuscation." arXiv preprint arXiv:2512.05379 (2025).

[4] Binder, Felix J., et al. "Looking inward: Language models can learn about themselves by introspection." arXiv preprint arXiv:2410.13787 (2024).

**Limitations:**

Overall, they discuss some practical limitations. However, if the larger models are not utilized, they should discuss if capability can limit the detection.

**Strengths And Weaknesses:**

Soundness: The paper is overall sound, but can benefit from a clearer narrative.

Presentation: The overall figures and presentation are good. However, it would be better to make it clear which methods are novel, which are reused, and what may be a hybrid. In Table 1, the methods can be labelled to say which is theirs.

Significance: The significance of the work is limited due to the limited number of models evaluated. The authors use a small model, Mistral-7B, compared to the most used models in the LLM research space. Moreover, a wider range of models (with more focus on popular/stronger) should be used to generate the machine/paraphrasing content.

Originality: The originality spans two niche directions: generating content to be more human-like and the finding that providing more content restores stylistic fingerprints. However, the findings are challenged by the limited models evaluated and the baseline of simple prompting with stronger models.

---

> ### Author Rebuttal · Authors · 2026-03-31
>
> > The significance of the work is limited due to the limited number of models evaluated. [...] a wider range of models (with more focus on popular/stronger) should be used to generate the machine/paraphrasing content.
>
> We clarify that while our *paraphraser* is Mistral-7B, the *machine text being transformed* comes from gpt-4o-mini, Llama-3-8B-Instruct, and Mistral-7B-Instruct (Section 5.2).
> That said, we have added three newer generators, Qwen3-8B, Qwen3-14B, and Mistral-Nemo (12B), and repeated the experiment in Section 6.1.
> We find that our core finding holds, namely that our style-aware paraphraser remains effective at reducing detectability across all tested detectors.
> These tables show that our paraphraser effectively re-writes the text of even these stronger generators so as to evade all detectors effectively:
>
> ### New Models Average (Qwen3-8B, Qwen3-14B, Mistral-Nemo) — Reddit (Max AUROC @ 1% FPR)
> |          |      1 |      5 |     10 |     25 |     50 |
> |:---------|-------:|-------:|-------:|-------:|-------:|
> | Baseline | 0.8653 | 0.9995 | 1.0000 | 1.0000 | 1.0000 |
> | Ours     | 0.5452 | 0.6075 | 0.6760 | 0.7803 | 0.8816 |
>
>
> ### New Models Average (Qwen3-8B, Qwen3-14B, Mistral-Nemo) — Amazon (Max AUROC @ 1% FPR)
> |          |      1 |      5 |     10 |     25 |     50 |
> |:---------|-------:|-------:|-------:|-------:|-------:|
> | Baseline | 0.9049 | 0.9999 | 1.0000 | 1.0000 | 1.0000 |
> | Ours     | 0.5232 | 0.5747 | 0.6213 | 0.7888 | 0.8373 |
>
>
> ### New Models Average (Qwen3-8B, Qwen3-14B, Mistral-Nemo) — Blogs (Max AUROC @ 1% FPR)
> |          |      1 |      5 |     10 |     25 |     50 |
> |:---------|-------:|-------:|-------:|-------:|-------:|
> | Baseline | 0.9074 | 0.9998 | 1.0000 | 1.0000 | 1.0000 |
> | Ours     | 0.5904 | 0.8675 | 0.9764 | 1.0000 | 1.0000 |
>
> Regarding scaling the paraphraser itself: our method deliberately uses a 7B model to demonstrate that style-aware paraphrasing is effective even at modest scale.
> A stronger paraphraser would likely improve results further, which reinforces rather than undermine our conclusions, since the quality of the generated paraphrases would likely increase.
>
> > The originality spans two niche directions [...] the findings are challenged by the limited models evaluated and the baseline of simple prompting with stronger models.
>
> We hope the new models address this.
> We also note that we evaluated against three prompting baselines (Section 5.1), including gpt-4o-mini, arguably the largest model in our evaluation.
> One of the baselines, OUTFOX, is a few-shot prompting baseline that incorporates in-context examples of text detected as human or machine by StyleDetect when re-writing text for evasion.
> Even against this strong baseline, the stylistic features remain robust.
>
> > It would be better to make it clear which methods are novel, which are reused, and what may be a hybrid. In Table 1, the methods can be labelled to say which is theirs.
>
> We will add clear labels to Table 1 distinguishing our contributions from baselines.
>
> > Q1: [...] Can the authors test their proposed stylistic change and the discovery about few-shot examples with these as baselines?
>
> Firstly, we would like to note that our OUTFOX baseline is itself an LLM-based detector that uses few-shot examples to detect machine-generated text.
> However, following your suggestion, we also tested Qwen3-14B as a zero-shot and few-shot detector.
> It performs at chance in both settings, motivating both the use of dedicated detectors and further reinforcing our conclusion that our style-aware paraphraser is an effective attack:
>
> |                   | Zero-Shot | Few-Shot |
> |-------------------|----------:|---------:|
> | Human vs. Machine |    0.4983 |   0.5019 |
> | Human vs. Ours    |    0.5001 |   0.5022 |
>
>
> > Q2: How does the detection change when you use larger models in your StyleDetect pipeline?
>
> We interpret this as asking whether StyleDetect remains effective when detecting text from larger or stronger LLMs.
> We tested StyleDetect on text from the new generators in both matched and cross-model settings (rows = support model, columns = target generator).
> StyleDetect remains robust even when support and target differ:
>
> ### AUROC@FPR=1% for StyleDetect
> | Support \ Target | Qwen3-8B | Qwen3-14B | Mistral-Nemo |
> |------------------|----------|-----------|---------------|
> | Qwen3-8B         | 0.87     | 0.91      | 0.75          |
> | Qwen3-14B        | 0.86     | 0.92      | 0.76          |
> | Mistral-Nemo     | 0.82     | 0.88      | 0.81          |
>
> > Q3: Any thoughts about quality after paraphrasing? The Table 8 example has one 'Ours' example that is in all caps; is this mimicking a specific human-author style?
>
> Yes, exactly.
> That example is mimicking a specific human author who sometimes writes in all caps.
>
> We will add all new results to the camera-ready version of the paper.

---

> > ### Author Rebuttal · Reviewer_GGRk · 2026-03-31
> >
> > I think the authors have acknowledged and answered the questions. After careful reading, I would like to stick to my original score and push for acceptance. I would encourage authors to discuss additional models as future work for Q1, as Qwen3-14B is smaller and less capable.

---

### Decision · Program_Chairs · 2026-04-30

**Decision:**

Accept (regular)

**Comment:**

The paper is generally sound and well presented, with clear motivation and a coherent integration of style-conditional paraphrasing, DPO-based optimization, and iterative reasoning. The figures and writing are strong and effectively support the arguments, and the work offers an interesting shift from surface-level detection toward stylistic feature spaces and multi-document analysis. However, the presentation would benefit from a clearer distinction between novel contributions and prior or hybrid methods, and Table 1 should explicitly indicate which components are newly proposed by the authors. The main limitation is the relatively narrow experimental scope, with heavy reliance on Mistral-7B and a limited set of baselines, which constrains the strength and generalizability of the conclusions. While the method achieves promising state-of-the-art performance under single-sample detection and demonstrates the robustness of stylistic fingerprints against paraphrasing-based attacks, its overall impact is somewhat limited by the restricted model coverage and partially outdated evaluation settings. Based on its merits and strengths, I support weak acceptance.